# Understanding Linear Probing then Fine-tuning Language Models from NTK Perspective

**Akiyoshi Tomihari**
The University of Tokyo
tomihari@g.ecc.u-tokyo.ac.jp

**Issei Sato**
The University of Tokyo
sato@g.ecc.u-tokyo.ac.jp

## Abstract

The two-stage fine-tuning (FT) method, linear probing (LP) then fine-tuning (LP-FT), outperforms linear probing and FT alone. This holds true for both in-distribution (ID) and out-of-distribution (OOD) data. One key reason for its success is the preservation of pre-trained features, achieved by obtaining a near-optimal linear head during LP. However, despite the widespread use of large language models, there has been limited exploration of more complex architectures such as Transformers. In this paper, we analyze the training dynamics of LP-FT for classification tasks on the basis of the neural tangent kernel (NTK) theory. Our analysis decomposes the NTK matrix into two components. This decomposition highlights the importance of the linear head norm alongside the prediction accuracy at the start of the FT stage. We also observe a significant increase in the linear head norm during LP, which stems from training with the cross-entropy (CE) loss. This increase in the linear head norm effectively reduces changes in learned features. Furthermore, we find that this increased norm can adversely affect model calibration, which can be corrected using temperature scaling. Additionally, we extend our analysis with the NTK to the low-rank adaptation (LoRA) method and validate its effectiveness. Our experiments using a Transformer-based model on multiple natural language processing datasets confirm our theoretical analysis. Our study demonstrates the effectiveness of LP-FT for fine-tuning language models. Code is available at https://github.com/tom4649/lp-ft_ntk.

## 1 Introduction

Fine-tuning pre-trained models for new tasks is a common practice across various fields. However, simply fine-tuning the entire model can lead to overfitting on training data, which may negatively impact generalization and out-of-distribution (OOD) performance [Li et al., 2020, Lee et al., 2023]. To address this, the two-stage approach known as linear probing then fine-tuning (LP-FT) [Kumar et al., 2022] has demonstrated high performance on both in-distribution (ID) and OOD data. Initially, linear probing (LP) optimizes only the linear head of the model, after which fine-tuning (FT) updates the entire model, including the feature extractor and the linear head. This method has been extensively analyzed and enhanced [Trivedi et al., 2023, Ren et al., 2023, Ha et al., 2024, Kirichenko et al., 2023].

The feature distortion theory, introduced by Kumar et al. [2022], explains the effectiveness of LP-FT on the basis of a theoretical analysis with a two-layer linear model. This theory suggests that LP-FT minimizes changes to pre-trained features by starting FT with an already optimized linear head from LP. However, our understanding of LP-FT, particularly when applied to complex architectures such as Transformers [Vaswani et al., 2017], remains incomplete. Thus, it is crucial to further explore the training dynamics of LP-FT in more complex models than the two-layer linear model.

In this paper, we apply the neural tangent kernel (NTK) theory [Jacot et al., 2018] to clarify the mechanisms underlying LP-FT, focusing on the training dynamics of classification models. The NTK is a theoretical tool that analyzes training dynamics by applying a first-order approximation

to changes in the model outputs with respect to its parameters. Therefore, the NTK is suited for analyzing feature changes during FT dynamics [Wei et al., 2022, Malladi et al., 2023]. Our analysis reveals that after LP, both more accurate predictions and increased norms of the linear head compared to their initial values contribute to minimizing feature changes. We then identify a significant increase in the linear head norm during LP from the analysis of training with cross-entropy (CE) loss, which contributes to small feature changes in the FT stage. On the other hand, we found that this increase in the linear head norm can worsen calibration, causing predicted probabilities to deviate from actual probabilities, which can be corrected with temperature scaling [Guo et al., 2017]. Furthermore, we extend our analysis based on the NTK to the low-rank adaptation (LoRA) method [Hu et al., 2022], a parameter-efficient fine-tuning strategy, and validate its effectiveness.

Our contributions are summarized as follows:

- We show that both accurate predictions and increased norms of the linear head during LP reduce feature changes in LP-FT within the NTK regime (Section 4), which is consistent with the feature distortion theory. (Corollary 4.3).
- We find that norms of the linear head significantly affect the balance of the NTK matrix components and influence the training dynamics of FT (Proposition 4.1).
- We also highlight that increased linear head norms can negatively affect model calibration, and this can be fixed with temperature scaling.
- We extend our analysis based on the NTK to the LoRA method and provide a theoretical validation of its efficacy (Proposition 4.4).

## 2   Related work

**LP-FT**   FT and LP are well-established transfer learning techniques with extensive empirical and theoretical studies [Zhuang et al., 2020, Kornblith et al., 2019, Tripuraneni et al., 2020]. Kumar et al. [2022] analyzed the effectiveness of these techniques using a two-layer linear model. Then, they proposed LP-FT that is a combined approach of LP then FT. Building on this study, subsequent studies have explored LP-FT in more detail. Trivedi et al. [2023] investigated LP-FT through the lens of safety objectives, proposing modifications to mitigate simplicity bias. Ren et al. [2023] analyzed LP-FT from the perspective of the initial discrepancy between predicted and actual probabilities, emphasizing the importance of the number of probing epochs during LP. Ha et al. [2024] further improved LP-FT by aligning batch normalization layers with the target domain. Kirichenko et al. [2023] highlighted the challenge that models depend on spurious features and proposed last-layer retraining as a cost-effective strategy to improve model robustness.

**Other FT methods**   Various FT strategies other than LP-FT have been proposed, including two-stage approaches [Zhang et al., 2020], regularization-based techniques [Jiang et al., 2019], and parameter-efficient fine-tuning methods [Houlsby et al., 2019, He et al., 2022]. One prominent example of a parameter-efficient method is LoRA, proposed by Hu et al. [2022]. This approach draws inspiration from the concept of intrinsic dimensions [Aghajanyan et al., 2021], suggesting that data can be effectively represented in a lower-dimensional space. Zeng and Lee [2024] explored the expressive power of LoRA, and Jang et al. [2024] provided a theoretical analysis of its convergence properties. However, challenges remain in parameter-efficient FT methods, including potential instability issues identified by Chen et al. [2022].

**Neural tangent kernel (NTK)**   The NTK, which was first introduced by Jacot et al. [2018], has become a valuable tool for analyzing the training dynamics of neural networks. Studies by Lee et al. [2019] and Arora et al. [2019] used the NTK to gain insights into how networks learn. Building on this foundation, Wei et al. [2022] introduced the concept of the empirical NTK, which extends the application of NTK to FT scenarios. This approach replaces the randomly initialized parameters in the standard NTK with the parameters of the pre-trained models. Further expanding on the empirical NTK, Malladi et al. [2023] conducted a theoretical and experimental investigation and found that prompt-based fine-tuning exhibits behavior consistent with the predictions of the kernel framework. Jang et al. [2024] extended this perspective to analyze LoRA.

## 3 Preliminary

In this section, we provide an overview of the FT methods used in this paper, followed by a brief explanation of the NTK.

**LP-FT**    In standard FT, the parameters of the linear head, weight $V$ and bias $b$, are initialized with random values. In contrast, in LP-FT, LP is conducted before the FT stage, and the FT stage is started with the obtained parameters. The performance of LP-FT is higher than that of LP and FT on both ID and OOD data [Kumar et al., 2022]. The original LP-FT paper [Kumar et al., 2022] explains the reason behind it as the feature distortion theory: the success of LP-FT stems from the minimal feature changes because of starting the FT stage with the linear head parameters which are close to the optimal solution. We analyze the training process of LP-FT throughout this paper.

**LoRA**    LoRA [Hu et al., 2022] introduces trainable rank decomposition matrices into each layer of the Transformer architecture. This approach, inspired by the concept of "intrinsic dimensions" from Aghajanyan et al. [2021], constrains updates to pre-trained weight matrices via low-rank decomposition. The update of a pre-trained weight matrix $W_0 \in \mathbb{R}^{q \times s}$ is approximated by $W + \Delta W = W_0 + B^{\text{LoRA}} A^{\text{LoRA}}$, where $B^{\text{LoRA}} \in \mathbb{R}^{q \times r}$ and $A^{\text{LoRA}} \in \mathbb{R}^{r \times s}$ are the only matrices optimized during fine-tuning. Here, $r \ll \min(q, s)$ represents the small intrinsic rank of the weight matrix, reflecting the low-rank approximation. The standard initialization of $B^{\text{LoRA}}$ and $A^{\text{LoRA}}$ is $B^{\text{LoRA}} = O$ and $A^{\text{LoRA}}$ is drawn from a normal distribution with mean 0.

**Neural tangent kernel (NTK)**    Jacot et al. [2018] introduced the NTK, which captures the training dynamics over time. They demonstrated that in the infinite width limit, the NTK remains constant. In this limit, training dynamics are governed by a linear model derived from a first-order Taylor expansion around the initial parameters of the network, known as the linearized or NTK regime [Lee et al., 2019]. For networks with finite width, this limiting kernel depends on the initialization parameters and is known as the empirical NTK [Wei et al., 2022]. Although the empirical NTK differs from the infinite width limit, it is valuable for analyzing the local training dynamics of models [Ren et al., 2022, Fort et al., 2020, Mohamadi and Sutherland, 2023, Wei et al., 2022, Jang et al., 2024], and has been used in FT [Ren et al., 2023, Malladi et al., 2023].

## 4 Analysis of LP-FT from NTK perspective

The original analysis of LP-FT by Kumar et al. [2022] is based on a two-layer linear model and proposes the feature distortion theory, which suggests that minimal changes in pre-trained features are the reason behind the robust performance of LP-FT. In this section, we use the NTK theory to analyze LP-FT to better understand the training dynamics of LP-FT in complex models like Transformers. After introducing the notation, we discuss the increase in the classifier weight norm during training, followed by the training dynamics in the NTK regime. We then extend our analysis to the LoRA method. These analyses suggest the LP-FT reduces feature distortion with the increased norm of the classifier weight and the near-optimal prediction after LP.

### 4.1 Notation

Let $\mathcal{X} = \{x_1, \ldots, x_N\} \subseteq \mathbb{R}^d$ represent the training samples, paired with labels from the set $\mathcal{Y} = \{y_1, \ldots, y_N\} \subseteq \{1, 2, \ldots, C\}$, where $d$, $C$, and $N$ denote the dimensions of the input space, the number of classes, and the number of training samples, respectively. This forms a training dataset $\{(x_1, y_1), \ldots, (x_N, y_N) \mid x_i \in \mathcal{X}, y_i \in \mathcal{Y}\}$, and we use $x \in \mathbb{R}^d$ to denote both a training and a test sample. We denote the $k$-th element of vector $a$ as $[a]_k$. We use the Euclidean norm $\| \cdot \|$ for vectors and the Frobenius norm $\| \cdot \|_F$ for matrices. $\langle \cdot, \cdot \rangle$ denotes the inner product of two vectors. $e_k$ represents the one-hot vector for class $k$, and $I_C$ is the identity matrix of size $C$.

The model function, denoted as $f(\cdot; \theta) : \mathcal{X} \to \mathbb{R}^C$, is parameterized by a set of parameters $\theta$, and sometimes abbreviated as $f(\cdot)$. The model includes a linear head, also referred to as the classifier, which consists of a weight matrix $V$ and a bias vector $b$. The feature extractor is denoted by $\phi(\cdot) : \mathbb{R}^h \to \mathbb{R}^C$, where $h$ represents the hidden dimension. The output of the model is given by $f(x) = V\phi(x) + b$. Parameters for a function $g(\cdot)$ and matrix $A$ are sometimes denoted as $\theta^g$ and $\theta^A$, respectively. Subscripts represent iteration or epoch, so $f_t(\cdot)$ denotes the model at time $t$.

With the loss function $\ell : \mathbb{R}^C \times \mathcal{Y} \to \mathbb{R}$, the training objective is to minimize the empirical risk $L(\boldsymbol{f}) := L(\boldsymbol{f}(\cdot; \theta)) = \frac{1}{N} \sum_{i=1}^{N} \ell(\boldsymbol{f}(\boldsymbol{x}_i; \theta), y_i)$. We use the CE loss, $\ell(\boldsymbol{f}(\boldsymbol{x}), y) = -\log([\boldsymbol{\sigma}_{\mathrm{SM}}(\boldsymbol{f}(\boldsymbol{x}))]_y)$, where $\boldsymbol{\sigma}_{\mathrm{SM}} : \mathbb{R}^C \to \mathbb{R}^C$ is the softmax function with its $k$-th element given by $[\boldsymbol{\sigma}_{\mathrm{SM}}(\boldsymbol{f}(\boldsymbol{x}))]_k = \frac{\exp([\boldsymbol{f}(\boldsymbol{x})]_k)}{\sum_{k'} \exp([\boldsymbol{f}(\boldsymbol{x})]_{k'})}$.

## 4.2 Training dynamics in the NTK regime

We use the NTK [Jacot et al., 2018], more specifically the empirical NTK [Wei et al., 2022, Malladi et al., 2023], to analyze the training dynamics of both FT and LP-FT. The empirical NTK, defined as the NTK with the parameters at the start of training, is a valuable tool for understanding the neural network training process, particularly in the context of FT [Wei et al., 2022, Malladi et al., 2023, Ren et al., 2023]. The empirical NTK applies a first-order approximation to changes in model outputs with respect to its parameters, so this is expected to capture changes in features.

To investigate the feature distortion theory in FT and LP-FT, we decomposed the updates into the following two parts. The part influenced by feature updates, unique to FT and absent in LP, is termed the *FT-effective* component of the NTK matrix, represented as $\boldsymbol{F}(\boldsymbol{x}, \boldsymbol{x}_i)$. In contrast, the part not influenced by feature updates, common to both FT and LP, determined by the pre-trained model, is termed the *pre-train-effective* component, represented as $\boldsymbol{P}(\boldsymbol{x}, \boldsymbol{x}_i)$. This decomposition highlights the distinct training dynamics of LP-FT in the NTK regime in the following proposition.

**Proposition 4.1** (FT in the NTK regime). *The NTK of a model $\boldsymbol{f}(\boldsymbol{x}) = \boldsymbol{V}\boldsymbol{\phi}(\boldsymbol{x}) + \boldsymbol{b}$, denoted by $\Theta^{\boldsymbol{f}}$, can be decomposed as:*

$$\Theta^{\boldsymbol{f}}(\boldsymbol{x}, \boldsymbol{x}_i) = \boldsymbol{P}(\boldsymbol{x}, \boldsymbol{x}_i) + \boldsymbol{F}(\boldsymbol{x}, \boldsymbol{x}_i),$$

*where the pre-train-effective component $\boldsymbol{P}(\boldsymbol{x}, \boldsymbol{x}_i)$ and the FT-effective component $\boldsymbol{F}(\boldsymbol{x}, \boldsymbol{x}_i)$ are defined using the classifier weight matrix $\boldsymbol{V}_0$ and the feature extractor $\boldsymbol{\phi}_0$ at starting point of training as:*

$$\boldsymbol{P}(\boldsymbol{x}, \boldsymbol{x}_i) := (\langle \boldsymbol{\phi}_0(\boldsymbol{x}), \boldsymbol{\phi}_0(\boldsymbol{x}_i) \rangle + 1)\boldsymbol{I}_C,$$

$$\boldsymbol{F}(\boldsymbol{x}, \boldsymbol{x}_i) := \boldsymbol{V}_0 \frac{\partial \boldsymbol{\phi}_0(\boldsymbol{x})}{\partial \theta^{\phi}} \frac{\partial \boldsymbol{\phi}_0(\boldsymbol{x}_i)}{\partial \theta^{\phi}}^{\top} \boldsymbol{V}_0^{\top}.$$

*Consequently, assuming that one-epoch training within the NTK regime approximates FT, the logits and feature vectors for a sample $\boldsymbol{x}$ after FT, denoted as $\boldsymbol{f}^{FT}(\boldsymbol{x})$ and $\boldsymbol{\phi}^{FT}(\boldsymbol{x})$, to the starting point of training, $\boldsymbol{f}_0(\boldsymbol{x})$ and $\boldsymbol{\phi}_0(\boldsymbol{x})$, can be expressed as:*

$$\boldsymbol{f}^{FT}(\boldsymbol{x}) - \boldsymbol{f}_0(\boldsymbol{x}) = \eta \sum_{i=1}^{N} (\boldsymbol{P}(\boldsymbol{x}, \boldsymbol{x}_i) + \boldsymbol{F}(\boldsymbol{x}, \boldsymbol{x}_i)) \, \boldsymbol{\delta}_i, \tag{1}$$

$$\boldsymbol{\phi}^{FT}(\boldsymbol{x}) - \boldsymbol{\phi}_0(\boldsymbol{x}) = \eta \sum_{i=1}^{N} \Theta^{\phi}(\boldsymbol{x}, \boldsymbol{x}_i) \boldsymbol{V}_0^{\top} \boldsymbol{\delta}_i, \tag{2}$$

*where $\Theta^{\phi}$ is the NTK matrix of the feature extractor $\phi$, $\boldsymbol{\delta}_i := \boldsymbol{e}_{y_i} - \boldsymbol{\sigma}_{SM}(\boldsymbol{f}_0(\boldsymbol{x}_i))$ represents the difference between the one-hot label for the class $y_i$ and the predicted probability, and $\eta$ is the learning rate.*

The proof of this proposition is included in the Appendix (Appendix A.2.1). In our decomposition of the NTK matrix, the pre-train-effective component $\boldsymbol{P}(\boldsymbol{x}, \boldsymbol{x}_i)$ is a diagonal matrix and remains unchanged after LP, while the FT-effective component $\boldsymbol{F}(\boldsymbol{x}, \boldsymbol{x}_i)$ is not a diagonal matrix and does change after LP, resulting in distinct characteristics for these components. The Frobenius norm of the classifier weight matrix, $\|\boldsymbol{V}_0\|_F$, influences the balance between the pre-train-effective and FT-effective components because it affects only the FT-effective component. This indicates that the classifier weight norm $\|\boldsymbol{V}_0\|_F$ has a significant impact on the training dynamics of FT.

**Hypothesis on reduced feature changes in LP-FT**     The above proposition provides insights into why LP-FT causes fewer feature changes compared to FT:

1. The impact of the classifier weight norm $\|\boldsymbol{V}_0\|_F$ differs in the equations: it affects feature changes linearly (2) and affects logits quadratically (1). This implies that a higher norm can result in significant logit updates with relatively minor changes to the feature extractor, reducing feature changes in LP-FT compared with FT due to the increased classifier weight norm after LP.

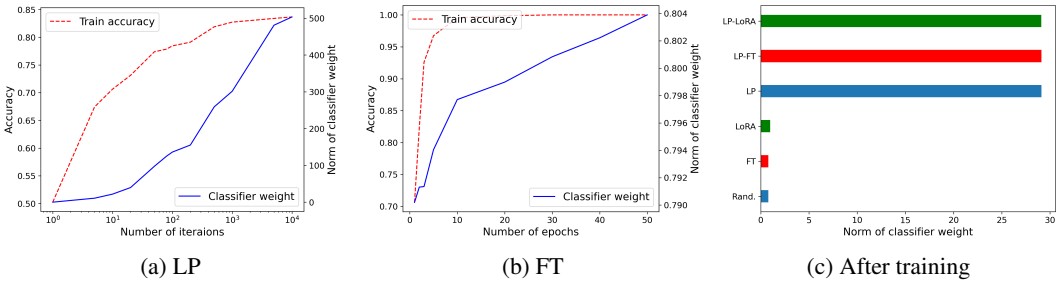

(a) LP             (b) FT             (c) After training

Figure 1: Increase in classifier weight norms during training on the RTE dataset. (a) and (b) show the increase of the both accuracy and classifier weight norms with training. (c) shows classifier weights norms after training.

2. The magnitude of changes in both features and logits ((1) and (2)), is proportional to $\delta_i$, the difference between the predicted probability and the one-hot label. This suggests that feature changes are less pronounced in LP-FT than in FT since the difference $\delta_i$ is smaller after LP.

3. The learning rate $\eta$, typically smaller in LP-FT than in FT [Kumar et al., 2022, Ren et al., 2023, Ha et al., 2024], helps moderate the direct influence of large classifier weight norms.

Prior studies [Kumar et al., 2022, Ren et al., 2023] have suggested that reduced feature changes in LP-FT stem from the near-optimal linear head obtained during LP. However, our analysis reveals that feature changes in LP-FT are also influenced by the classifier weight norm $V_0$ after LP. Our analysis focusing on classifier weight norms provides a new perspective on the training dynamics of LP-FT, highlighting the importance of the classifier weight norm in reducing feature distortion.

### 4.3 Derivation of Lemma A.3 from Kumar et al. in the NTK regime

The analysis presented in the original LP-FT paper by Kumar et al. [Kumar et al., 2022] operates within a framework where the feature extractor is a linear function. We define this framework in our context as follows:

**Definition 4.2** (Linear model [Kumar et al., 2022]). *A linear model is defined as* $f_{\text{linear}}(x) := VBx + b$, *where* $V \in \mathbb{R}^{C \times h}$ *is the classifier weight matrix and* $B \in \mathbb{R}^{h \times d}$ *is the weight matrix of the feature extractor.*

The linear model is a model whose feature extractor $\phi$ is a linear transformation. In this setting, we derive a corollary from Proposition 4.1 in our context, which is the pivotal lemma in the original LP-FT analysis [Kumar et al., 2022]:

**Corollary 4.3** (Lemma A.3 from Kumar et al. in the NTK regime). *Within the context of the linear model (Definition 4.2), for any sample* $x \in \text{Span}(\mathcal{X})^{\perp}$, *the orthogonal complement of the subspace spanned by the training sample set* $\mathcal{X}$, *the features after FT remain unchanged, expressed as:*

$$\phi^{FT}(x) = \phi_0(x),$$

*where* $\phi^{FT}(x)$ *and* $\phi_0(x)$ *denote the feature vectors after and before FT, respectively.*

This corollary shows that feature vectors for the samples in the orthogonal complement of training sample subspace are not updated. Therefore, given that pre-trained features have characteristics beneficial to downstream tasks, significant feature changes in FT, dependent on small training samples in LP, lead to poor generalization and OOD performance. The proof of this lemma can be found in the Appendix (Appendix A.2.2).

### 4.4 Increase in the classifier weight norm

The analysis in the previous section suggests that the classifier weight norm affects both feature changes and logits. On the basis of this insight, we examine classifier weight norms during training. Figure 1 shows that classifier weight norms consistently increase over time for LP, standard FT, and LoRA. As the training proceeds, norms of classifier bias and logits increases, while training loss decreases. Notably, LP shows a significantly larger increase in the norm compared to FT and LoRA.

Consider the transpose of the $k$-th row of matrix $\boldsymbol{V}$ denoted as $\boldsymbol{v}_k \in \mathbb{R}^h$ for $1 \leq k \leq C$, where $C$ is the number of classes. Let $\tau_{ki}$ represent the angle between $\boldsymbol{\phi}(\boldsymbol{x_i})$ and $\boldsymbol{v}_k$, which expands $\langle \boldsymbol{v}_k, \boldsymbol{\phi}(\boldsymbol{x_i}) \rangle$ to $\|\boldsymbol{v}_k\|\|\boldsymbol{\phi}(\boldsymbol{x_i})\| \cos \tau_{ki}$. The probability that class $k$ is chosen for sample $\boldsymbol{x_i}$ is given by the softmax function $[\boldsymbol{\sigma}_{\mathrm{SM}}(\boldsymbol{f}(\boldsymbol{x_i}))]_k = \frac{\exp(\langle \boldsymbol{v}_k, \boldsymbol{\phi}(\boldsymbol{x_i}) \rangle)}{\sum_{k'} \exp(\langle \boldsymbol{v}_{k'}, \boldsymbol{\phi}(\boldsymbol{x_i}) \rangle)}$. Consequently, with the CE loss for an input $\boldsymbol{x_i}$ classified into class $y_i$ defined as $\ell(\boldsymbol{f}(\boldsymbol{x_i}), y_i) = -\log([\boldsymbol{\sigma}_{\mathrm{SM}}(\boldsymbol{f}(\boldsymbol{x_i}))]_{y_i})$, we have the following partial derivatives:

$$\frac{\partial \ell(\boldsymbol{f}(\boldsymbol{x_i}), y_i)}{\partial \cos \tau_{ki}} = \begin{cases} [\boldsymbol{\sigma}_{\mathrm{SM}}(\boldsymbol{f}(\boldsymbol{x_i}))]_k \|\boldsymbol{v}_k\|\|\boldsymbol{\phi}(\boldsymbol{x_i})\| & \text{if } k \neq y_i, \\ -(1 - [\boldsymbol{\sigma}_{\mathrm{SM}}(\boldsymbol{f}(\boldsymbol{x_i}))]_{y_i})\|\boldsymbol{v}_{y_i}\|\|\boldsymbol{\phi}(\boldsymbol{x_i})\| & \text{if } k = y_i, \end{cases}$$

where the derivative with respect to $\cos \tau_{y_i i}$ is negative and positive for $k \neq y_i$. As training progresses, $\cos \tau_{y_i i}$ tends to increase towards positivity, while $\cos \tau_{ki}$ for $k \neq y_i$ tends to become negative for each $i$. The derivative with respect to $\|\boldsymbol{v}_k\|$ is given by:

$$\frac{\partial L(\boldsymbol{f})}{\partial \|\boldsymbol{v}_k\|} = \sum_{i=1}^{N} \left( \sum_{k \neq y_i} [\boldsymbol{\sigma}_{\mathrm{SM}}(\boldsymbol{f}(\boldsymbol{x_i}))]_k \|\boldsymbol{\phi}(\boldsymbol{x_i})\| \cos \tau_{ki} - \sum_{k=y_i} (1 - [\boldsymbol{\sigma}_{\mathrm{SM}}(\boldsymbol{f}(\boldsymbol{x_i}))]_{y_i})\|\boldsymbol{\phi}(\boldsymbol{x_i})\| \cos \tau_{y_i i} \right).$$
(3)

Therefore, with adequate training and $\cos \tau_{ki} < 0$ and $\cos \tau_{y_i i} > 0$, the derivative with respect to $\|\boldsymbol{v}_k\|$ is likely to become negative for each class $k$. The training of the model proceeds so that the empirical risk $L$ decreases, so the norm $\|\boldsymbol{v}_k\|$ tends to increase. This finding aligns with prior studies [Soudry et al., 2018, Kim and Kim, 2020].

**Remark: increase in classifier weight norms is more pronounced in LP than in FT**  In FT, particularly within an overparameterized setting, the model $\boldsymbol{f}$ may achieve perfect classification on the training dataset. That is, $[\boldsymbol{\sigma}_{\mathrm{SM}}(\boldsymbol{f}(\boldsymbol{x_i}))]_k$ becomes close to 0 for $k \neq y_i$ and 1 for $k = y_i$. In this scenario, the derivative in Eq. (3) becomes close to zero, or the training itself is finished. Conversely, perfect classification is typically unattainable in LP unless the training dataset is linearly separable, so the derivative continues to be negative. In addition, while all parameters are updated in FT, only the classifier is optimized in LP, so the change in the classifier weight needs to be larger in LP than in FT to achieve the same classification performance. Consequently, the classifier weight norm tends to increase more significantly in LP than in FT, as shown in Figure 1 (c).

## 4.5   Training process of LoRA

We extend our analysis based on the NTK to the training process of LoRA. We follow the linear model setting as in Definition 4.2 and analyze the training dynamics of LoRA in the NTK regime.

**Proposition 4.4** (LoRA approximates FT). *Consider the linear model setting (Definition 4.2) and let $\boldsymbol{f}^{LoRA}$ and $\boldsymbol{f}^{FT}$ be the models obtained via one-epoch training with LoRA and standard FT in the NTK regime. Let $r$ denote the rank of the LoRA hyperparameter, and $\sigma^2$ represent the variance of the low-rank weight matrix initialization. Assume the input samples $\boldsymbol{x}$ satisfy $\|\boldsymbol{x}\| \leq c$. Then, for each sample pair $\boldsymbol{x}_i, \boldsymbol{x}_j \in \mathcal{X}$, the pre-train-effective components of the NTK matrix for LoRA and FT, $\boldsymbol{P}^{LoRA}(\boldsymbol{x}_i, \boldsymbol{x}_j)$ and $\boldsymbol{P}^{FT}(\boldsymbol{x}_i, \boldsymbol{x}_j)$, are identical:*

$$\boldsymbol{P}^{LoRA}(\boldsymbol{x}_i, \boldsymbol{x}_j) = \boldsymbol{P}^{FT}(\boldsymbol{x}_i, \boldsymbol{x}_j).$$

*Moreover, with at least $1 - 4\exp(-(\epsilon^2 - \epsilon^3)r/4)$ probability, their FT-effective components, $\boldsymbol{F}^{LoRA}(\boldsymbol{x}_i, \boldsymbol{x}_j)$ and $\boldsymbol{F}^{FT}(\boldsymbol{x}_i, \boldsymbol{x}_j)$, satisfy:*

$$\|\boldsymbol{F}^{LoRA}(\boldsymbol{x}_i, \boldsymbol{x}_j) - \sigma^2 r \boldsymbol{F}^{FT}(\boldsymbol{x}_i, \boldsymbol{x}_j)\| \leq c\epsilon \|\boldsymbol{V}_0 \boldsymbol{V}_0^\top\|.$$

This proposition suggests that with high probability, the only difference of the NTK matrix between LoRA and standard FT is a scalar factor of the FT-effective component in the NTK matrix, and the scalar factor depends on the hyperparameters of LoRA. This implies that when the hyperparameters of LoRA are set appropriately, LoRA training is similar to standard FT training. This is consistent with the analysis by Malladi et al. [2023], where the NTK matrix of LoRA and standard FT are close to each other. It is important to note that the proposition is also valid for LP-FT and LP-LoRA (LP then LoRA). The proof of this proposition is included in the Appendix (Appendix A.2.3).

## 4.6 Discussion

An increased norm of the classifier weight reduces feature distortion and enhances the contribution of the FT-effective component of the NTK matrix during training. As a result, a higher classifier weight norm in LP-FT can be advantageous. However, since the increased norm is dependent on LP training, its optimality is not guaranteed. Specifically, during test time, although the increased classifier weight norm does not influence accuracy, it affects the calibration of the model. Calibration is defined as the alignment between the predicted probabilities and the actual probabilities [Guo et al., 2017]. An excessively high classifier weight norm can lead to overconfident predictions, which might be detrimental in practical applications. Consequently, there is potential for refining LP-FT by adjusting the classifier weight norm to enhance calibration.

Tuning the norm of the classifier after training can be effectively equated to applying temperature scaling [Guo et al., 2017] at test time. Temperature scaling adjusts the output logits with a temperature parameter $T$, thereby improving model calibration. Specifically, temperature scaling with parameter $T$, expressed as $\boldsymbol{f}(\boldsymbol{x})/T = \frac{\boldsymbol{V}}{T}\phi(\boldsymbol{x}) + \frac{\boldsymbol{b}}{T}$, can be viewed as scaling the norm of classifier weight $\boldsymbol{V}$ and bias $\boldsymbol{b}$ by the temperature parameter $T$.

# 5 Numerical evaluation with transformer models

In this section, we numerically justify the following aspects obtained from our analysis:

- The changes in features during training are smaller in LP-FT than in FT, and the norms of the classifier significantly increase during LP (Section 5.2).

- The FT-effective component of the NTK matrix more effectively captures the input data than the pre-train-effective component (Section 5.3) and is more pronounced in LP-FT than FT.

- A large classifier weight norm reduces the feature change during training, and its negative effects on calibration can be improved by temperature scaling (Section 5.4).

Details on the datasets, setup, and additional results, including performance evaluations for the experimental and practical application, are available in the Appendix (Appendices A.3 and A.4).

## 5.1 Setup

**Datasets and models** We used a total of 13 classification datasets from various benchmarks: SuperGLUE [Wang et al., 2019], GLUE [Wang et al., 2018], BOSS [Yuan et al., 2023], and PubMed 20k RCT [Dernoncourt and Lee, 2017]. The breakdown of the datasets is as follows: five datasets from SuperGLUE (BoolQ, CB, RTE, WiC, and WSC), three datasets from GLUE (CoLA, MRPC, and SST-2), four datasets from BOSS (Amazon, Dynasent, SemEval, and SST-5), and PubMed 20k RCT. Following experimental settings in studies that analyze FT dynamics from NTK perspectives [Malladi et al., 2023, Jang et al., 2024] and the study with similar settings Chen et al. [2022], we employed the RoBERTa-base model [Liu et al., 2020] as our Transformer-based model.

Table 1: Changes in features (F) and classifier (C) norms on the CB and RTE datasets. CS, Diff, FDR, and Norm represent the cosine similarity between features, the difference in norms from the pre-trained model, Fisher's discriminant ratio, and the norm, respectively. After LP-FT, Diff(F) is smaller compared to FT, while preserving the high CS(F) and low FDR(F) of the pre-trained features. In contrast, Norm(C) is significantly larger after LP and LP-FT than both the pre-trained model and after FT. This trend is also observed when training with LoRA.

| Method | CB | | | | RTE | | | |
|---|---|---|---|---|---|---|---|---|
| | CS(F) | Diff(F) | FDR(F) | Norm(C) | CS(F) | Diff(F) | FDR(F) | Norm(C) |
| Pre-trained | 0.997 | — | $8.14 \times 10^4$ | $9.51 \times 10^{-1}$ | 0.996 | — | $8.59 \times 10^1$ | $7.76 \times 10^{-1}$ |
| LP | 0.997 | — | $8.14 \times 10^4$ | $2.48 \times 10^1$ | 0.996 | — | $8.59 \times 10^1$ | $3.10 \times 10^1$ |
| FT | 0.336 | $2.21 \times 10^1$ | $7.39 \times 10^8$ | $9.60 \times 10^{-1}$ | 0.260 | $2.16 \times 10^1$ | $1.42 \times 10^4$ | $7.84 \times 10^{-1}$ |
| LoRA | 0.499 | $1.92 \times 10^1$ | $8.91 \times 10^6$ | $1.43 \times 10^0$ | 0.759 | $1.06 \times 10^1$ | $2.97 \times 10^3$ | $1.21 \times 10^0$ |
| LP-FT | 0.804 | $1.20 \times 10^1$ | $6.47 \times 10^6$ | $2.48 \times 10^1$ | 0.942 | $4.70 \times 10^0$ | $1.57 \times 10^2$ | $3.10 \times 10^1$ |
| LP-LoRA | 0.837 | $9.08 \times 10^0$ | $2.10 \times 10^6$ | $2.49 \times 10^1$ | 0.924 | $4.63 \times 10^0$ | $2.06 \times 10^1$ | $3.10 \times 10^1$ |

Table 2: Kernel statistics on the CB dataset. FN, Acc, and FT Ratio denote the Frobenius norm, kernel regression accuracy, and contribution of the FT-effective component, respectively. Pre-train E and FT E refer to the pre-train-effective and FT-effective components of the NTK matrix.

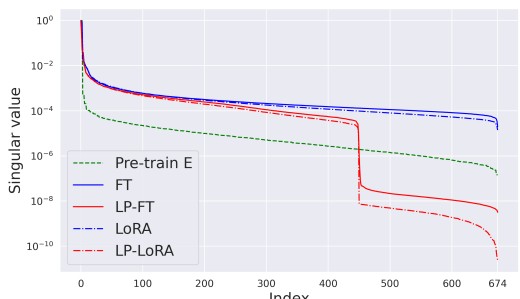

| Method | Kernel | Rank | FN($\times 10^3$) | Acc (train/test) | FT Ratio |
|---|---|---|---|---|---|
| - | Pre-train E | 18 | 51.0 | 87.11/79.17 | - |
| FT | FT E | 608 | 13.9 | 84.74/79.76 | |
| | NTK | 210 | 64.9 | 84.74/79.76 | 0.1987 |
| LoRA | FT E | 500 | 0.0226 | 86.22/79.17 | |
| | NTK | 20 | 51.0 | 92.15/84.52 | 0.0004 |
| LP-FT | FT E | 344 | 7250 | 100.00/86.31 | |
| | NTK | 344 | 7280 | 100.00/86.31 | 1.0000 |
| LP-LoRA | FT E | 307 | 15.1 | 94.96/85.71 | |
| | NTK | 188 | 62.6 | 95.11/85.71 | 1.0137 |

Figure 2: Singular value distribution normalized by the maximum value on the CB dataset, showing the common pre-train-effective component (Pre-train E) and the FT-effective components for each training option.

**Implementation and training** We used the Transformers library [Wolf et al., 2020] and Adapter-Hub [Pfeiffer et al., 2020] for our implementation. Our training protocol followed the experimental setup described by Chen et al. [2022]. Hyperparameter tuning, especially for learning rates during the FT stage of LP-FT, was conducted through a grid search based on the validation set performance. For LP, we used logistic regression with L2 regularization on pre-trained features.

## 5.2 Small feature changes during LP-FT and significant norm increase during LP

LP-FT achieves notable performance with Transformer-based language models, outperforming standard FT in both ID and OOD settings, as detailed in Appendix (Appendices A.4.1 and A.4.3). To understand the underlying reasons for these results and validate small feature changes suggested by our analysis (Section 4.2), we analyzed changes in both the classifier and the features.

According to statistics presented in Table 1, the features after LP-FT demonstrate smaller changes from those of the pre-trained model than FT. Consequently, LP-FT preserves high cosine similarity among its features and exhibits a low Fisher's discriminant ratio (FDR) [Fisher, 1936], which assesses linear separability. Conversely, the classifier norms after LP and LP-FT are substantially larger than those of the pre-trained model and after FT, suggesting a significant increase in classifier weights during LP. A similar trend is observed in training with LoRA.

## 5.3 Kernel analysis

We examined the overall NTK matrix and its pre-train-effective and FT-effective components to understand their properties. Kernel regression was performed on the train and test sets to evaluate the performance of each kernel matrix.

**Analysis of NTK matrix components and effectiveness of LP-FT** In Table 2, the FT-effective component of the NTK matrix for LP-FT shows a higher rank and greater kernel regression accuracy compared to the pre-train-effective component, and the overall NTK matrix has intermediate properties. Additionally, the FT-effective component contributes more significantly to the overall kernel in LP-FT than in FT, as indicated by a higher FT Ratio. This ratio, calculated as the average of $\| \sum_{i=1}^{N} \boldsymbol{F}(\boldsymbol{x}, \boldsymbol{x}_i)\boldsymbol{\delta}_i\|/\| \sum_{i=1}^{N} \left( \boldsymbol{P}(\boldsymbol{x}, \boldsymbol{x}_i) + \boldsymbol{F}(\boldsymbol{x}, \boldsymbol{x}_i) \right) \boldsymbol{\delta}_i \|$ for the train set samples, reflects the enhanced influence of the FT-effective component in LP-FT than in FT. These results suggest that the NTK matrix of LP-FT better captures input data through the increased influence of the FT-effective component.

**Similarities between LoRA and FT** The ranks of the FT-effective components in LoRA and FT (or LP-LoRA and LP-FT) are similar, as indicated in Table 2. Their distributions of singular values normalized by the maximum singular value, also closely align, as shown in Figure 2. These results suggest that the FT-effective components of the NTK matrix in FT and LoRA differ only by a scalar factor. This consistency demonstrates that our analysis (Section 4.2), originally based on a two-layer linear model, is applicable to more complex Transformer-based models.

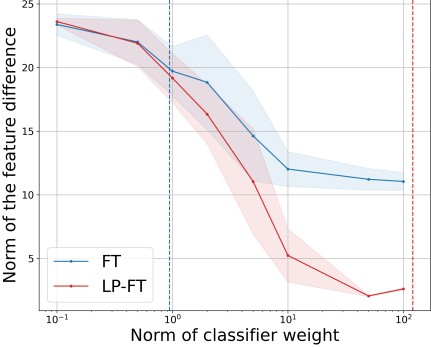

Figure 3: Feature differences on SST-5 (OOD). Solid lines show mean values; shaded areas represent standard errors. Dashed vertical lines indicate the classifier weight norm after training. This figure validates our analysis that larger classifier weight norms reduce feature changes.

Table 3: ECE and MCE with temperature scaling on the test set of the RTE dataset. w/o TS and w/ TS denote without and with temperature scaling, respectively, and Imp. represents the improvement because of temperature scaling. We bold the best improvements. This table shows that poor calibration of LP-FT can be effectively mitigated through temperature scaling.

| Metric | Method | w/o TS | w/ TS | Imp. |
|--------|--------|--------|-------|------|
| ECE (%) | FT | 21.16 | 5.13 | 16.03 |
| | LP-FT | 21.72 | 5.48 | **16.24** |
| | LoRA | 11.92 | 6.17 | 5.76 |
| | LP-LoRA | 18.14 | 5.72 | 12.42 |
| MCE (%) | FT | 53.11 | 25.87 | 27.24 |
| | LP-FT | 63.95 | 13.94 | **50.01** |
| | LoRA | 25.04 | 13.75 | 11.29 |
| | LP-LoRA | 40.46 | 18.82 | 21.63 |

## 5.4 Analysis of classifier weight norms and temperature scaling

We experimentally verified significant effects of classifier weight norms in training (Section 4.2) and at test time (Section 4.6) in the following.

**Effects of classifier weight norms in training** We scaled the classifier weight norms at the start of the FT stage of LP-FT. The results, shown in Figure 3, indicate that larger classifier weight norms almost monotonically lead to smaller feature differences in both FT and LP-FT. Notably, LP-FT consistently shows smaller feature differences than FT, particularly when the classifier weight norms are large, validating our analysis that larger classifier weight norms reduce feature changes.

**Temperature scaling at test time** We implemented temperature scaling at test time, which is equivalent to adjusting the classifier weight norms, as discussed in Section 4.6. We optimized the temperature parameters on the validation sets based on CE loss, following the methodology suggested by Guo et al. [2017]. Table 3 presents the results on the RTE datasets. We assessed the expected calibration error (ECE) and maximum calibration error (MCE) [Naeini et al., 2015], which quantify the absolute differences between predicted and actual probabilities, with lower values indicating better calibration. These results show that the improvements in calibration with temperature scaling are the largest in LP-FT for both ECE and MCE, with notably substantial improvements in MCE. This suggests that large classifier weight norms contribute to poor calibration of LP-FT, which can be effectively mitigated through temperature scaling. These results highlight the effectiveness of refining LP-FT by temperature scaling.

## 6 Conclusion

In this paper, we explored the LP-FT training dynamics in complex classification models using the NTK to analyze feature changes. Our analysis identified classifier weight norms at the start of the FT stage as a key factor influencing FT dynamics. These norms balance the NTK matrix components and help reduce feature changes. Our findings support the existing feature distortion theory from an NTK perspective and emphasize the role of classifier weight norms alongside prediction accuracy. We also found that increases in classifier weight norms, characteristic of training with CE loss, may negatively impact model calibration, and this can be mitigated by temperature scaling. Additionally, the approximation effectiveness of LoRA is theoretically validated in terms of the similarity of the NTK matrix components. Empirical experiments with Transformer-based language models supported our theoretical insights, validating our understanding of the NTK, feature changes, and the benefits of temperature scaling. Overall, our study substantiates the efficacy of LP-FT as a robust method for adapting pre-trained complex models while preserving their well-trained features.

**Limitations**    The main limitation of our study is that it is based on the NTK regime, which might not fully capture the training dynamics. Additionally, we consider just one epoch of gradient descent in FT, which may not effectively represent the overall training. In our experiments, we specifically focused on validating the effectiveness of LP-FT on language models. Therefore, areas other than natural language processing are outside the scope of our experiments.

## Acknowledgments and Disclosure of Funding

This work was supported by JSPS KAKENHI Grant Number 24H00709 Japan.

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

# A    Appendix / supplemental material

## A.1    Abbreviation and notation

Table 4 and Table 5 show our abbreviations and notations, respectively.

Table 4: Table of abbreviations.

| Abbreviation | Definition |
|---|---|
| FT | fine-tuning |
| LP | linear probing |
| LP-FT | linear probing then fine-tuning |
| NTK | neural tangent kernel |
| LoRA | low rank adaptation [Hu et al., 2022] |
| ECE | expected calibration error ([Naeini et al., 2015]) |
| MCE | maximum calibration error (Naeini et al. [2015]) |
| ID / OOD | in-distribution / out-of-distribution |
| FDR | Fisher's discriminant ratio [Fisher, 1936] |

## A.2    Proof of theoretical results

**Additional notation**   The parameters for a function $\boldsymbol{g}$, a weight matrix $\boldsymbol{A}$, and a vector $\boldsymbol{a}$ is denoted as $\theta^{\boldsymbol{g}}, \theta^{\boldsymbol{A}}$, and $\theta^{\boldsymbol{a}}$. Given a function $\boldsymbol{g}(\cdot; \theta^{\boldsymbol{g}}) : \mathbb{R}^d \to \mathbb{R}^s$ trained on $N$ training samples $\mathcal{X} = \{\boldsymbol{x}_1, \boldsymbol{x}_2, \ldots, \boldsymbol{x}_N\} \subseteq \mathbb{R}^d$, we denote the NTK matrix of $\boldsymbol{g}$ at time $t$ as $\Theta_t^{\boldsymbol{g}}$, which is defined as $\Theta_t^{\boldsymbol{g}} := \frac{\partial \boldsymbol{g}_t(\mathcal{X})}{\partial \theta^{\boldsymbol{g}}} \left( \frac{\partial \boldsymbol{g}_t(\mathcal{X})}{\partial \theta^{\boldsymbol{g}}} \right)^\top \in \mathbb{R}^{Ns \times Ns}$, where $\boldsymbol{g}_t(\mathcal{X}) := \text{vec}\left( \boldsymbol{g}_t(\boldsymbol{x}_i) \right)_{\boldsymbol{x}_i \in \mathcal{X}}$, and $\frac{\partial \boldsymbol{g}_t(\mathcal{X})}{\partial \theta^{\boldsymbol{g}}} \in \mathbb{R}^{Ns \times p}$ with $p$ parameters. The sub-matrix $\Theta_t^{\boldsymbol{g}}(\boldsymbol{x}_i, \boldsymbol{x}_j)$ is defined as $\Theta_t^{\boldsymbol{g}}(\boldsymbol{x}_i, \boldsymbol{x}_j) := \frac{\partial \boldsymbol{g}_t(\boldsymbol{x}_i)}{\partial \theta^{\boldsymbol{g}}} \left( \frac{\partial \boldsymbol{g}_t(\boldsymbol{x}_j)}{\partial \theta^{\boldsymbol{g}}} \right)^\top \in \mathbb{R}^{s \times s}$, describing the relationship between training samples $\boldsymbol{x}_i$ and $\boldsymbol{x}_j$ in $\mathcal{X}$. In the infinite width limit with NTK parameterization and general assumptions, the NTK matrix converges to $\Theta^{\boldsymbol{g}} := \frac{\partial \boldsymbol{g}_0(\mathcal{X})}{\partial \theta^{\boldsymbol{g}}} \left( \frac{\partial \boldsymbol{g}_0(\mathcal{X})}{\partial \theta^{\boldsymbol{g}}} \right)^\top$ as shown by [Jacot et al., 2018]. Subscripts represent iteration or epoch, so $\boldsymbol{g}_t(\cdot)$ denotes the model $\boldsymbol{g}$ at time $t$. $\otimes$ denotes the kronecker product of two matrices defined as

$$\boldsymbol{A} \otimes \boldsymbol{B} := \begin{bmatrix} a_{11}\boldsymbol{B} & a_{12}\boldsymbol{B} & \cdots & a_{1n}\boldsymbol{B} \\ a_{21}\boldsymbol{B} & a_{22}\boldsymbol{B} & \cdots & a_{2n}\boldsymbol{B} \\ \vdots & \vdots & \ddots & \vdots \\ a_{m1}\boldsymbol{B} & a_{m2}\boldsymbol{B} & \cdots & a_{mn}\boldsymbol{B} \end{bmatrix},$$

where $\boldsymbol{A} = [a_{ij}]$ is an $m \times n$ matrix and $\boldsymbol{B}$ is any matrix.

Table 5: Table of notations.

| Variable | Definition |
| --- | --- |
| $C$ / $N$ | number of classes / training samples |
| $d$ / $h$ / $r$ | input dimension / hidden dimension / rank of LoRA |
| $\mathcal{X}$ / $\mathcal{Y}$ | trainig samples / labels |
| $\boldsymbol{x}$ / $y$ | sample / label |
| $[\boldsymbol{a}]_k$ | $k$-th element of vector $\boldsymbol{a}$ |
| $\|\cdot\|$ / $\|\cdot\|_F$ / $\langle\cdot,\cdot\rangle$ | Euclidean norm / Frobenius norm / inner product |
| $\boldsymbol{e}_y$ | one-hot encoding of label $y$ |
| $\boldsymbol{I}_C$ | $C \times C$ identity matrix |
| $\ell(\boldsymbol{f}(\boldsymbol{x}), y)$ | loss function |
| $L$ | empirical risk |
| $\boldsymbol{\sigma}_{\text{SM}}$ | softmax function |
| $\boldsymbol{f}(\boldsymbol{x})$ | model output |
| $\boldsymbol{\phi}(\boldsymbol{x})$ | feature extractor |
| $\boldsymbol{V}$ / $\boldsymbol{b}$ | classifier weight / bias |
| $\boldsymbol{V}_0$ / $\boldsymbol{\phi}_0$ | classifier weight / feature extractor at the start of training |
| $\boldsymbol{B}$ | feature extractor weight matrix in two-layer linear model |
| $\boldsymbol{A}^{\text{LoRA}}$ / $\boldsymbol{B}^{\text{LoRA}}$ | low-rank weight matrices in LoRA |
| $\theta^g$ / $\theta^A$ / $\theta^a$ | parameter of function $\boldsymbol{g}$ / matrix $\boldsymbol{A}$ / vector $\boldsymbol{a}$ |
| $\Theta^{\boldsymbol{f}}$ / $\Theta^{\boldsymbol{\phi}}$ | NTK matrix of model / feature extractor |
| $\boldsymbol{P}(\boldsymbol{x}, \boldsymbol{x}_i)$ / $\boldsymbol{F}(\boldsymbol{x}, \boldsymbol{x}_i)$ | pre-train-effective / FT-effective component of NTK matrix |
| $\boldsymbol{\delta}_i$ | difference between one-hot label and predicted probability |
| $\eta$ | learning rate |
| $\otimes$ | kronecker product of two matrices |

### A.2.1 Proof of Proposition 4.1

**Proposition 4.1.** The NTK matrix of a model $\boldsymbol{f}(\boldsymbol{x}) = \boldsymbol{V}\boldsymbol{\phi}(\boldsymbol{x}) + \boldsymbol{b}$, denoted by $\Theta^{\boldsymbol{f}}$, can be decomposed as:

$$\Theta^{\boldsymbol{f}}(\boldsymbol{x}, \boldsymbol{x}_i) = \boldsymbol{P}(\boldsymbol{x}, \boldsymbol{x}_i) + \boldsymbol{F}(\boldsymbol{x}, \boldsymbol{x}_i),$$

where the pre-train-effective component $\boldsymbol{P}(\boldsymbol{x}, \boldsymbol{x}_i)$ and the FT-effective component $\boldsymbol{F}(\boldsymbol{x}, \boldsymbol{x}_i)$ are defined using the classifier weight matrix $\boldsymbol{V}_0$ and the feature extractor $\boldsymbol{\phi}_0$ at starting point of training as:

$$\boldsymbol{P}(\boldsymbol{x}, \boldsymbol{x}_i) := (\langle\boldsymbol{\phi}_0(\boldsymbol{x}), \boldsymbol{\phi}_0(\boldsymbol{x}_i)\rangle + 1)\boldsymbol{I}_C,$$

$$\boldsymbol{F}(\boldsymbol{x}, \boldsymbol{x}_i) := \boldsymbol{V}_0 \frac{\partial\boldsymbol{\phi}_0(\boldsymbol{x})}{\partial\theta^{\phi}} \frac{\partial\boldsymbol{\phi}_0(\boldsymbol{x}_i)}{\partial\theta^{\phi}}^{\top} \boldsymbol{V}_0^{\top}.$$

Consequently, assuming that one-epoch training within the NTK regime approximates FT, the logits and feature vectors for a sample $\boldsymbol{x}$ after FT, denoted as $\boldsymbol{f}^{\text{FT}}(\boldsymbol{x})$ and $\boldsymbol{\phi}^{\text{FT}}(\boldsymbol{x})$, to the starting point of training, $\boldsymbol{f}_0(\boldsymbol{x})$ and $\boldsymbol{\phi}_0(\boldsymbol{x})$, can be expressed as:

$$\boldsymbol{f}^{\text{FT}}(\boldsymbol{x}) - \boldsymbol{f}_0(\boldsymbol{x}) = \eta \sum_{i=1}^{N} (\boldsymbol{P}(\boldsymbol{x}, \boldsymbol{x}_i) + \boldsymbol{F}(\boldsymbol{x}, \boldsymbol{x}_i)) \boldsymbol{\delta}_i,$$

$$\boldsymbol{\phi}^{\text{FT}}(\boldsymbol{x}) - \boldsymbol{\phi}_0(\boldsymbol{x}) = \eta \sum_{i=1}^{N} \Theta^{\phi}(\boldsymbol{x}, \boldsymbol{x}_i) \boldsymbol{V}_0^{\top} \boldsymbol{\delta}_i,$$

where $\boldsymbol{\delta}_i := \boldsymbol{e}_{y_i} - \boldsymbol{\sigma}_{\text{SM}}(\boldsymbol{f}_0(\boldsymbol{x}_i))$ represents the difference between the one-hot label and the predicted probability, and $\eta$ is the learning rate.

**Proof of Proposition 4.1**

*Proof.* The parameters of $\boldsymbol{f}$, denoted as $\boldsymbol{\theta^f}$, consist of $\boldsymbol{\theta^V}$, $\boldsymbol{\theta^b}$, and $\boldsymbol{\theta^\phi}$. The derivative of the model $\boldsymbol{f}$ with respect to each parameter is given by:

$$\frac{\partial \boldsymbol{f}(\boldsymbol{x})}{\partial \boldsymbol{\theta^V}} = \boldsymbol{\phi}(\boldsymbol{x})^\top \otimes \boldsymbol{I}_C, \tag{4}$$

$$\frac{\partial \boldsymbol{f}(\boldsymbol{x})}{\partial \boldsymbol{\theta^b}} = \boldsymbol{I}_C, \tag{5}$$

$$\frac{\partial \boldsymbol{f}(\boldsymbol{x})}{\partial \boldsymbol{\theta^\phi}} = \boldsymbol{V}\frac{\partial \boldsymbol{\phi}(\boldsymbol{x})}{\partial \boldsymbol{\theta^\phi}}. \tag{6}$$

Therefore, the NTK matrix of $\boldsymbol{f}$, defined as $\Theta^{\boldsymbol{f}}(\boldsymbol{x}, \boldsymbol{x}_i) := \frac{\partial \boldsymbol{f}_0(\boldsymbol{x})}{\partial \boldsymbol{\theta^f}}\left(\frac{\partial \boldsymbol{f}_0(\boldsymbol{x}_i)}{\partial \boldsymbol{\theta^f}}\right)^\top$, can be expressed as:

$$
\begin{aligned}
\Theta^{\boldsymbol{f}}(\boldsymbol{x}, \boldsymbol{x}_i) &= \frac{\partial \boldsymbol{f}_0(\boldsymbol{x})}{\partial \boldsymbol{\theta^f}}\left(\frac{\partial \boldsymbol{f}_0(\boldsymbol{x}_i)}{\partial \boldsymbol{\theta^f}}\right)^\top \\
&= \frac{\partial \boldsymbol{f}_0(\boldsymbol{x})}{\partial \boldsymbol{\theta^V}}\frac{\partial \boldsymbol{f}_0(\boldsymbol{x}_i)}{\partial \boldsymbol{\theta^V}}^\top + \frac{\partial \boldsymbol{f}_0(\boldsymbol{x})}{\partial \boldsymbol{\theta^b}}\frac{\partial \boldsymbol{f}_0(\boldsymbol{x}_i)}{\partial \boldsymbol{\theta^b}}^\top + \frac{\partial \boldsymbol{f}_0(\boldsymbol{x})}{\partial \boldsymbol{\theta^\phi}}\frac{\partial \boldsymbol{f}_0(\boldsymbol{x}_i)}{\partial \boldsymbol{\theta^\phi}}^\top \\
&= \left(\boldsymbol{\phi}_0(\boldsymbol{x})^\top \otimes \boldsymbol{I}_C\right)\left(\boldsymbol{\phi}_0(\boldsymbol{x}_i)^\top \otimes \boldsymbol{I}_C\right)^\top + \boldsymbol{I}_C + \boldsymbol{V}\frac{\partial \boldsymbol{\phi}_0(\boldsymbol{x})}{\partial \boldsymbol{\theta^\phi}}\left(\boldsymbol{V}\frac{\partial \boldsymbol{\phi}_0(\boldsymbol{x}_i)}{\partial \boldsymbol{\theta^\phi}}\right)^\top \quad (\because \text{Eqs. } (4), (5), (6)) \\
&= \langle \boldsymbol{\phi}_0(\boldsymbol{x}), \boldsymbol{\phi}_0(\boldsymbol{x}_i)\rangle \boldsymbol{I}_C + \boldsymbol{I}_C + \boldsymbol{V}_0\frac{\partial \boldsymbol{\phi}_0(\boldsymbol{x})}{\partial \boldsymbol{\theta^\phi}}\left(\frac{\partial \boldsymbol{\phi}_0(\boldsymbol{x}_i)}{\partial \boldsymbol{\theta^\phi}}\right)^\top \boldsymbol{V}_0^\top \\
&= (\langle \boldsymbol{\phi}_0(\boldsymbol{x}), \boldsymbol{\phi}_0(\boldsymbol{x}_i)\rangle + 1)\boldsymbol{I}_C + \boldsymbol{V}_0\frac{\partial \boldsymbol{\phi}_0(\boldsymbol{x})}{\partial \boldsymbol{\theta^\phi}}\left(\frac{\partial \boldsymbol{\phi}_0(\boldsymbol{x}_i)}{\partial \boldsymbol{\theta^\phi}}\right)^\top \boldsymbol{V}_0^\top \\
&= \boldsymbol{P}(\boldsymbol{x}, \boldsymbol{x}_i) + \boldsymbol{F}(\boldsymbol{x}, \boldsymbol{x}_i). \tag{7}
\end{aligned}
$$

For gradient descent, the update to the parameters $\boldsymbol{\theta^f}$ at time $t$ is given by:

$$
\begin{aligned}
\boldsymbol{\theta}^{\boldsymbol{f}}_{t+1} - \boldsymbol{\theta}^{\boldsymbol{f}}_t &= -\eta\left(\frac{\partial L(\boldsymbol{f}_t)}{\partial \boldsymbol{\theta^f}}\right)^\top \\
&= \eta \sum_{i=1}^{N}\left(\frac{\partial \log([\boldsymbol{\sigma}_{\mathrm{SM}}(\boldsymbol{f}(\boldsymbol{x}_i))]_{y_i})}{\partial \boldsymbol{f}_t(\boldsymbol{x}_i)}\frac{\partial \boldsymbol{f}_t(\boldsymbol{x}_i)}{\partial \boldsymbol{\theta^f}}\right)^\top \\
&= \eta \sum_{i=1}^{N}\left((\boldsymbol{e}_{y_i} - \boldsymbol{\sigma}_{\mathrm{SM}}(\boldsymbol{f}(\boldsymbol{x}_i)))^\top \frac{\partial \boldsymbol{f}_t(\boldsymbol{x}_i)}{\partial \boldsymbol{\theta^f}}\right)^\top \\
&= \eta \sum_{i=1}^{N} \frac{\partial \boldsymbol{f}_t(\boldsymbol{x}_i)}{\partial \boldsymbol{\theta^f}}^\top \boldsymbol{\delta}_i, \tag{8}
\end{aligned}
$$

where $\boldsymbol{\delta}_i$ is defined as $\boldsymbol{\delta}_i := \boldsymbol{e}_{y_i} - \boldsymbol{\sigma}_{\mathrm{SM}}(\boldsymbol{f}_0(\boldsymbol{x}_i))$. Assuming that one-epoch training approximates FT, the model is expressed as $\boldsymbol{f}^{\mathrm{FT}} = \boldsymbol{f}_1$. Therefore, the update to the model $\boldsymbol{f}$ in the linearized regime is given by:

$$
\begin{aligned}
\boldsymbol{f}^{\mathrm{FT}}(\boldsymbol{x}) - \boldsymbol{f}_0(\boldsymbol{x}) &= \boldsymbol{f}_1(\boldsymbol{x}) - \boldsymbol{f}_0(\boldsymbol{x}) \quad (\because \text{one-epoch approximation of fine-tuning}) \\
&= \frac{\partial \boldsymbol{f}_0(\boldsymbol{x})}{\partial \boldsymbol{\theta^f}}(\boldsymbol{\theta}^{\boldsymbol{f}}_1 - \boldsymbol{\theta}^{\boldsymbol{f}}_0) \quad (\because \text{linearized regime}) \\
&= \eta \sum_{i=1}^{N} \frac{\partial \boldsymbol{f}_0(\boldsymbol{x})}{\partial \boldsymbol{\theta^f}}\left(\frac{\partial \boldsymbol{f}_0(\boldsymbol{x}_i)}{\partial \boldsymbol{\theta^f}}\right)^\top \boldsymbol{\delta}_i \quad (\because \text{Eq. } (8)) \\
&= \eta \sum_{i=1}^{N} (\boldsymbol{P}(\boldsymbol{x}, \boldsymbol{x}_i) + \boldsymbol{F}(\boldsymbol{x}, \boldsymbol{x}_i))\boldsymbol{\delta}_i. \quad (\because \text{Eq. } (7))
\end{aligned}
$$

Finally, replacing $\boldsymbol{\theta}^f$ with $\boldsymbol{\theta}^\phi$ in Eq. (8), the update to the parameters $\boldsymbol{\theta}^\phi$ at time $t$ is given by

$$\boldsymbol{\theta}_{t+1}^\phi - \boldsymbol{\theta}_t^\phi = \eta \sum_{i=1}^N \frac{\partial \boldsymbol{f}_t(\boldsymbol{x}_i)}{\partial \boldsymbol{\theta}^\phi}^\top \boldsymbol{\delta}_i. \tag{9}$$

Therefore, the update to the feature extractor after FT, given by $\phi^{\text{FT}} = \phi_1$ for the same assumption, is:

$$\begin{aligned} \phi^{\text{FT}}(\boldsymbol{x}) - \phi_0(\boldsymbol{x}) &= \phi_1(\boldsymbol{x}) - \phi_0(\boldsymbol{x}) \\ &= \frac{\partial \phi_0(\boldsymbol{x})}{\partial \boldsymbol{\theta}^\phi}(\boldsymbol{\theta}_1^\phi - \boldsymbol{\theta}_0^\phi) \quad (\because \text{linearized regime}) \\ &= \frac{\partial \phi_0(\boldsymbol{x})}{\partial \boldsymbol{\theta}^\phi} \eta \sum_{i=1}^N \left(\frac{\partial \boldsymbol{f}_0(\boldsymbol{x}_i)}{\partial \boldsymbol{\theta}^\phi}\right)^\top \boldsymbol{\delta}_i \quad (\because \text{Eq. (9)}) \\ &= \frac{\partial \phi_0(\boldsymbol{x})}{\partial \boldsymbol{\theta}^\phi} \eta \sum_{i=1}^N \left(\boldsymbol{V}_0 \frac{\partial \phi_0(\boldsymbol{x}_i)}{\partial \boldsymbol{\theta}^\phi}\right)^\top \boldsymbol{\delta}_i \quad (\because \text{Eq. (6)}) \\ &= \eta \sum_{i=1}^N \frac{\partial \phi_0(\boldsymbol{x})}{\partial \boldsymbol{\theta}^\phi} \left(\frac{\partial \phi_0(\boldsymbol{x}_i)}{\partial \boldsymbol{\theta}^\phi}\right)^\top \boldsymbol{V}_0^\top \boldsymbol{\delta}_i \\ &= \eta \sum_{i=1}^N \Theta^\phi(\boldsymbol{x}, \boldsymbol{x}_i) \boldsymbol{V}_0^\top \boldsymbol{\delta}_i. \end{aligned}$$

This completes the proof. $\qquad\qquad\square$

### A.2.2 Proof of Corollary 4.3

**Corollary 4.3.** Within the context of the linear model (Definition 4.2), for any sample $\boldsymbol{x} \in \text{Span}(\mathcal{X})^\perp$, the orthogonal complement of the subspace spanned by the training sample set $\mathcal{X}$, the features after FT remain unchanged, expressed as:

$$\phi^{\text{FT}}(\boldsymbol{x}) = \phi_0(\boldsymbol{x}),$$

where $\phi^{\text{FT}}(\boldsymbol{x})$ and $\phi_0(\boldsymbol{x})$ denote the feature vectors after and before FT, respectively.

**Proof of Corollary 4.3**

*Proof.* The feature extractor is given by $\phi(\boldsymbol{x}) = \boldsymbol{B}\boldsymbol{x}$, where $\boldsymbol{B}$ is the weight matrix. The derivative of the feature extractor with respect to the parameters $\boldsymbol{\theta}^\phi = \boldsymbol{\theta}^B$ is:

$$\frac{\partial \phi(\boldsymbol{x})}{\partial \boldsymbol{\theta}^\phi} = \frac{\partial \boldsymbol{B}\boldsymbol{x}}{\partial \boldsymbol{\theta}^B} = \boldsymbol{x} \otimes \boldsymbol{I}_h,$$

so the empirical NTK matrix of the feature extractor becomes:

$$\begin{aligned} \Theta^\phi(\boldsymbol{x_i}, \boldsymbol{x_j}) &:= \frac{\partial \phi_0(\boldsymbol{x_i})}{\partial \boldsymbol{\theta}^\phi} \frac{\partial \phi_0(\boldsymbol{x_j})}{\partial \boldsymbol{\theta}^\phi}^\top \\ &= \langle \boldsymbol{x_i}, \boldsymbol{x_j} \rangle \otimes \boldsymbol{I}_h \end{aligned}$$

where $\otimes$ denotes the kronecker product.

From the Proposition 4.1, the feature update is given by:

$$\begin{aligned} \phi^{\text{FT}}(\boldsymbol{x}) - \phi_0(\boldsymbol{x}) &= \eta \sum_{i=1}^N \Theta^\phi(\boldsymbol{x}, \boldsymbol{x}_i) \boldsymbol{V}_0^\top \boldsymbol{\delta}_i \\ &= \eta \sum_{i=1}^N \langle \boldsymbol{x}, \boldsymbol{x}_i \rangle \boldsymbol{V}_0^\top \boldsymbol{\delta}_i, \end{aligned}$$

where $\boldsymbol{\delta}_i = \boldsymbol{e}_{y_i} - \boldsymbol{\sigma}_{\text{SM}}(\boldsymbol{f}_0(\boldsymbol{x}_i))$, $\boldsymbol{V}_0$ is the classifier weight matrix at the start of training, and $\eta$ is the learning rate. For any sample $\boldsymbol{x} \in \text{Span}(\mathcal{X})^\perp$, $\langle \boldsymbol{x}, \boldsymbol{x}_i \rangle = 0$ for all $\boldsymbol{x}_i \in \mathcal{X}$, so the feature update is 0 for OOD samples, namely:

$$\phi^{\text{FT}}(\boldsymbol{x}) - \phi_0(\boldsymbol{x}) = 0.$$

This completes the proof. $\qquad\qquad\square$

### A.2.3 Proof of Proposition 4.4

**Proposition 4.4.** Consider the linear model setting (Definition 4.2) and let $\boldsymbol{f}^{\text{LoRA}}$ and $\boldsymbol{f}^{\text{FT}}$ be the models obtained via one-epoch training with LoRA and standard FT in the NTK regime. Let $r$ denote the rank of the LoRA hyperparameter, and $\sigma^2$ represent the variance of the low-rank weight matrix initialization. Assume the input samples $\boldsymbol{x}$ satisfy $\|\boldsymbol{x}\| \le c$. Then, for each sample pair $\boldsymbol{x}_i, \boldsymbol{x}_j \in \mathcal{X}$, the pre-train-effective components of the NTK matrix for LoRA and FT, $\boldsymbol{P}^{\text{LoRA}}(\boldsymbol{x}_i, \boldsymbol{x}_j)$ and $\boldsymbol{P}^{\text{FT}}(\boldsymbol{x}_i, \boldsymbol{x}_j)$, are identical:

$$\boldsymbol{P}^{\text{LoRA}}(\boldsymbol{x}_i, \boldsymbol{x}_j) = \boldsymbol{P}^{\text{FT}}(\boldsymbol{x}_i, \boldsymbol{x}_j).$$

Moreover, with at least $1 - 4\exp(-(\epsilon^2 - \epsilon^3)r/4)$ probability, their FT-effective components, $\boldsymbol{F}^{\text{LoRA}}(\boldsymbol{x}_i, \boldsymbol{x}_j)$ and $\boldsymbol{F}^{\text{FT}}(\boldsymbol{x}_i, \boldsymbol{x}_j)$, satisfy:

$$\|\boldsymbol{F}^{\text{LoRA}}(\boldsymbol{x}_i, \boldsymbol{x}_j) - \sigma^2 r \boldsymbol{F}^{\text{FT}}(\boldsymbol{x}_i, \boldsymbol{x}_j)\| \le c\epsilon \|\boldsymbol{V}_0 \boldsymbol{V}_0^\top\|.$$

**Proof Approach** To prove this theorem, we use a lemma from distributional properties:

**Lemma A.1** (Corollary of the distributional Johnson-Lindenstrauss Lemma). *Given vectors $\boldsymbol{u}, \boldsymbol{v} \in \mathbb{R}^d$ with $\|\boldsymbol{u}\|, \|\boldsymbol{v}\| \le c$, and a random matrix $\boldsymbol{A} \in \mathbb{R}^{k \times d}$ with i.i.d. entries from a distribution with mean 0 and variance 1, for any $\epsilon > 0$:*

$$\Pr\left[|(\boldsymbol{A}\boldsymbol{u})^\top(\boldsymbol{A}\boldsymbol{v}) - \boldsymbol{u}^\top \boldsymbol{v}| \ge c\epsilon\right] \le 4\exp\left(-(\epsilon^2 - \epsilon^3)k/4\right).$$

**Proof of Proposition 4.4**

*Proof.* The feature vector of LoRA is given by $\phi^{\text{LoRA}}(\boldsymbol{x}) = \boldsymbol{B}_0 \boldsymbol{x} + \boldsymbol{B}^{\text{LoRA}} \boldsymbol{A}^{\text{LoRA}} \boldsymbol{x}$, where pretrained feature weight matrix $\boldsymbol{B}_0$ is fixed during training, and $\boldsymbol{A}^{\text{LoRA}} \in \mathbb{R}^{r \times d}$ and $\boldsymbol{B}^{\text{LoRA}} \in \mathbb{R}^{h \times r}$ are low-rank weight matrices in LoRA. $\boldsymbol{A}^{\text{LoRA}}$ is initialized from a normal distribution with mean 0 and variance $\sigma^2$, while $\boldsymbol{B}^{\text{LoRA}}$ is initialized with zeros. The LoRA feature updates are represented as $\phi^{\text{LoRA}}(\boldsymbol{x}) = \boldsymbol{B}_0 \boldsymbol{x} + \boldsymbol{B}^{\text{LoRA}} \boldsymbol{A}^{\text{LoRA}} \boldsymbol{x}$, with $\boldsymbol{B}_0$ fixed during training.

The pre-train-effective components of LoRA and FT, denoted as $\boldsymbol{P}^{\text{LoRA}}(\boldsymbol{x}, \boldsymbol{x}_i)$ and $\boldsymbol{P}^{\text{FT}}(\boldsymbol{x}, \boldsymbol{x}_i)$ respectively, are defined as:

$$\boldsymbol{P}^{\text{LoRA}}(\boldsymbol{x}, \boldsymbol{x}_i) = (\langle \phi_0^{\text{LoRA}}(\boldsymbol{x}), \phi_0^{\text{LoRA}}(\boldsymbol{x}_i)\rangle + 1)\boldsymbol{I}_C,$$
$$\boldsymbol{P}^{\text{FT}}(\boldsymbol{x}, \boldsymbol{x}_i) = (\langle \phi_0^{\text{FT}}(\boldsymbol{x}), \phi_0^{\text{FT}}(\boldsymbol{x}_i)\rangle + 1)\boldsymbol{I}_C,$$

where $\boldsymbol{I}_C$ is the identity matrix of size $C$. These pre-train-effective components are identical since:

$$\phi_0^{\text{LoRA}}(\boldsymbol{x}) = \boldsymbol{B}_0 \boldsymbol{x} + \boldsymbol{B}_0^{\text{LoRA}} \boldsymbol{A}_0^{\text{LoRA}} \boldsymbol{x} = \boldsymbol{B}_0 \boldsymbol{x} = \phi_0^{\text{FT}}(\boldsymbol{x}),$$

for all $\boldsymbol{x} \in \mathcal{X}$ because $\boldsymbol{B}^{\text{LoRA}}$ is initialized as a zero matrix i.e. $\boldsymbol{B}_0^{\text{LoRA}} = \boldsymbol{O}$.

For the FT-effective component of the NTK matrix, consider the derivatives concerning LoRA parameters $\boldsymbol{B}^{\text{LoRA}}$ and $\boldsymbol{A}^{\text{LoRA}}$:

$$\frac{\partial \phi^{\text{LoRA}}(\boldsymbol{x})}{\partial \theta^{\boldsymbol{B}^{\text{LoRA}}}} = \boldsymbol{A}\boldsymbol{x} \otimes \boldsymbol{V},$$
$$\frac{\partial \phi^{\text{LoRA}}(\boldsymbol{x})}{\partial \theta^{\boldsymbol{A}^{\text{LoRA}}}} = \boldsymbol{x} \otimes \boldsymbol{V}\boldsymbol{B}^{\text{LoRA}} \boldsymbol{B}^{\text{LoRA}\top} \boldsymbol{V}^\top.$$

Here, $\theta^{\boldsymbol{B}^{\text{LoRA}}}$ and $\theta^{\boldsymbol{A}^{\text{LoRA}}}$ denote the parameters of $\boldsymbol{B}^{\text{LoRA}}$ and $\boldsymbol{A}^{\text{LoRA}}$, respectively.

The FT-effective component of the NTK matrix for LoRA, denoted as $\boldsymbol{F}^{\text{LoRA}}(\cdot, \cdot)$, is derived by combining these partial derivatives:

$$
\begin{aligned}
\boldsymbol{F}^{\text{LoRA}}(\boldsymbol{x}, \boldsymbol{x}_i) =& \boldsymbol{V}_0 \left( \frac{\partial \phi_0^{\text{LoRA}}(\boldsymbol{x})}{\partial \theta^{\boldsymbol{B}^{\text{LoRA}}}} \frac{\partial \phi_0^{\text{LoRA}}(\boldsymbol{x}_i)}{\partial \theta^{\boldsymbol{B}^{\text{LoRA}}}}^\top + \frac{\partial \phi_0^{\text{LoRA}}(\boldsymbol{x})}{\partial \theta^{\boldsymbol{A}^{\text{LoRA}}}} \frac{\partial \phi_0^{\text{LoRA}}(\boldsymbol{x}_i)}{\partial \theta^{\boldsymbol{A}^{\text{LoRA}}}}^\top \right) \boldsymbol{V}_0^\top \\
=& \boldsymbol{V}_0 \left( \langle \boldsymbol{A}_0^{\text{LoRA}} \boldsymbol{x}, \boldsymbol{A}_0^{\text{LoRA}} \boldsymbol{x}_i \rangle + \langle \boldsymbol{x}, \boldsymbol{x}_i \rangle \boldsymbol{B}_0^{\text{LoRA}} \boldsymbol{B}_0^{\text{LoRA}\top} \right) \boldsymbol{V}_0^\top \\
=& \langle \boldsymbol{A}_0^{\text{LoRA}} \boldsymbol{x}, \boldsymbol{A}_0^{\text{LoRA}} \boldsymbol{x}_i \rangle \boldsymbol{V}_0 \boldsymbol{V}_0^\top,
\end{aligned}
$$

where the last equality holds because $\boldsymbol{B}_0^{\text{LoRA}}$ is a zero matrix.

Similarly, the FT-effective component of the NTK matrix for standard FT, $\boldsymbol{F}^{\text{FT}}(\cdot, \cdot)$, is given by:

$$\boldsymbol{F}^{\text{FT}}(\boldsymbol{x}, \boldsymbol{x}_i) = \boldsymbol{V}_0 \left( \frac{\partial \phi_0^{\text{FT}}(\boldsymbol{x})}{\partial \theta_B} \frac{\partial \phi_0^{\text{FT}}(\boldsymbol{x}_i)}{\partial \theta_B}^\top \right) \boldsymbol{V}_0^\top$$
$$= \langle \boldsymbol{x}, \boldsymbol{x}_i \rangle \boldsymbol{V}_0 \boldsymbol{V}_0^\top.$$

Using the Johnson-Lindenstrauss lemma, with a probability of at least $1 - 4\exp(-(\epsilon^2 - \epsilon^3)r/4)$:

$$|\langle \boldsymbol{A}^{\text{LoRA}}\boldsymbol{x}, \boldsymbol{A}^{\text{LoRA}}\boldsymbol{x}_i \rangle - \sigma^2 r \langle \boldsymbol{x}, \boldsymbol{x}_i \rangle| \leq c\sigma^2 r\epsilon,$$

which implies:

$$\|\boldsymbol{F}^{\text{LoRA}}(\boldsymbol{x}, \boldsymbol{x}_i) - \sigma^2 r \boldsymbol{F}^{\text{FT}}(\boldsymbol{x}, \boldsymbol{x}_i)\| = \|\langle \boldsymbol{A}^{\text{LoRA}}\boldsymbol{x}, \boldsymbol{A}^{\text{LoRA}}\boldsymbol{x}_i \rangle \boldsymbol{V}_0 \boldsymbol{V}_0^\top - \sigma^2 r \langle \boldsymbol{x}, \boldsymbol{x}_i \rangle \boldsymbol{V}_0 \boldsymbol{V}_0^\top\|$$
$$\leq |\langle \boldsymbol{A}^{\text{LoRA}}\boldsymbol{x}, \boldsymbol{A}^{\text{LoRA}}\boldsymbol{x}_i \rangle - \sigma^2 r \langle \boldsymbol{x}, \boldsymbol{x}_i \rangle| \|\boldsymbol{V}_0 \boldsymbol{V}_0^\top\|$$
$$\leq c\sigma^2 r\epsilon \|\boldsymbol{V}_0 \boldsymbol{V}_0^\top\|.$$

This completes the proof. $\qquad\square$

### A.3 Experimental details

#### A.3.1 Datasets

From the SuperGLUE benchmark [Wang et al., 2019], we used the five datasets: BoolQ [Clark et al., 2019], CB (CommitmentBank) [De Marneffe et al., 2019], RTE (Recognizing Textual Entailment) [Dagan et al., 2005, Bar-Haim et al., 2006, Giampiccolo et al., 2007, Bentivogli et al., 2009], WiC (Words in Context) [Burstein et al., 2019], and WSC (Winograd Schema Challenge) [Levesque et al., 2012]. From the GLUE benchmark [Wang et al., 2018], we used the three datasets: CoLA (Corpus of Linguistic Acceptability) [Warstadt et al., 2019], MRPC (Microsoft Research Paraphrase Corpus) [Dolan and Brockett, 2005], and SST-2 (Stanford Sentiment Treebank, version 2) [Socher et al., 2013]. Four datasets from BOSS [Yuan et al., 2023] were used in OOD evaluation: Amazon Reviews [McAuley and Leskovec, 2013], Dynasent [Potts et al., 2021], SemEval [Nakov et al., 2016], and SST-5 [Socher et al., 2013]. Finally, we used the PubMed 20k RCT dataset [Dernoncourt and Lee, 2017] for validation in practical settings. The dataset statistics are detailed in Table 7.

For the datasets from the GLUE, SuperGLUE, and BOSS benchmarks, we divided the original training set using a 9:1 training-to-validation ratio, using the original validation set as the test set, in accordance with Chen et al. [2022]. For PubMed 20k RCT, we used the original training, validation, and test sets for their respective purposes.

#### A.3.2 Implementation and training details

When applying LoRA, LoRA was applied only to the query and value projection matrices of the attention mechanism in the Transformer architecture, following the approach described in the original paper by Hu et al. [2022]. The LoRA settings were fixed at $\alpha = 8$ and $r = 8$ for all experiments.

The model was trained for 10 epochs without early stopping, and the one showing the best performance on the validation set was chosen for further evaluation. We used the Adam optimizer [Kingma and Ba, 2017]. Our code is built on PyTorch [Paszke et al., 2019], using the HuggingFace Transformers library [Wolf et al., 2020] and AdapterHub [Pfeiffer et al., 2020]. All experiments were run on a single NVIDIA A100 GPU. The results reported are averages from 3 tuning seeds and 5 evaluation seeds.

For LP, cross-validation and automatic hyperparameter adjustment were used to find the optimal L2 regularization strength, using scikit-learn [Pedregosa et al., 2011] with its standard training parameters.

Details on the hyperparameters for our experiments can be found in Table 6.

### A.3.3 Details of each experiment

**Experiments on the GLUE and SuperGLUE benchmarks**  For the FT and LoRA methods, the learning rate and batch size were adopted from Chen et al. [2022], where these hyperparameters were optimized using grid search on the validation set. For LP-FT and LP-LoRA, batch size is fixed at 8 and we tuned the learning rate.

**Experiments on BOSS benchmark and the PubMed 20k RCT dataset**  For the experiments on BOSS benchmark and the PubMed 20k RCT dataset, we tuned the learning rate and batch size using grid search based on the validation set performance.

**Calculation of the NTK matrix**  We computed the NTK matrix for FT, LoRA, LP-FT, and LP-LoRA as specified in Eq. (1). We separately calculated the pre-train-effective and FT-effective components of the NTK matrix. Following the methodology by Malladi et al. [2024], we used functorch [Horace He, 2021] and forward-mode auto-differentiation [Novak et al., 2022] for these calculations. To reduce computational costs, we randomly selected 10% of the parameters from the word embedding matrix for derivative calculations. For datasets with more than 250 samples, we used a subset of 250 randomly selected samples to compute the NTK matrix.

**Solving the kernel regression**  Following the methodology described by Malladi et al. [2024], we treated each output logit independently in our kernel regression model. This method is based on the representer theorem, where the empirical risk minimizer is expressed as a linear combination of kernel features from the training data: $\boldsymbol{f}(\boldsymbol{x}) = \sum_{i=1}^{NC} \alpha_i \boldsymbol{K}(\boldsymbol{x}, x_i)$, with $\boldsymbol{K}$ representing the NTK matrix or its component for a training set of size $NC \times NC$. We solved this optimization using logistic regression with L2 regularization and used the resulting coefficients $\alpha_i$ to compute logits on the test set via its corresponding NTK matrix.

**Effects of classifier weight norms in training**  We scaled the norms of the classifiers within the range of $[0.1, 0.5, 1, 2, 5, 10, 50, 100]$ before proceeding to the FT stage of training, specifically after random initialization in FT and after LP training in LP-FT. We conducted this experiment using the CB and RTE datasets and Boss benchmark. We apply the LoRA method on the CB and RTE datasets. We averaged the results over 5 seeds for the CB and RTE datasets and 3 seeds for the Boss benchmark, plotting these with their standard deviations.

**Temperature scaling**  We applied temperature scaling [Guo et al., 2017] to the logits of the model at test time. Following the methodology of the original paper [Guo et al., 2017], we tuned the temperature parameter using the validation set to minimize the negative log-likelihood. For implementation, we employed the Adam optimizer [Kingma and Ba, 2017] with a learning rate of $1 \times 10^{-3}$, optimizing the temperature for $1 \times 10^5$ steps. We incorporated early stopping based on the negative log-likelihood, with a patience of 10 iterations starting from an initial temperature value of 1.0. The number of the bins to calculate ECE and MCE is set to 15.

### A.4 Additional experimental results

### A.4.1 Results on the SuperGLUE and GLUE benchmarks

Table 8 shows the test results for the SuperGLUE and GLUE benchmarks. We report accuracy and its standard deviation on the test sets, except for the CoLA dataset, which uses the Matthew's correlation coefficient for the performance metric.

Figure 4 shows the increase in the classifier weight norm during training on the CB dataset. With more iterations or epochs, there is a noticeable increase in both accuracy and the classifier weight norm.

Figure 5 and Figure 6 display t-SNE visualizations of the feature vectors from the CB dataset. After FT, the features are distinctly separated by class. In contrast, the classifier row vectors remain nearly identical to those of the pre-trained model. After LP-FT, the features retain the structure of the pre-trained model, but the classifier row vectors deviate from their initial state. A similar pattern is observed with the LoRA method.

Table 6: Hyperparameter configurations. The settings include batch size (bs), learning rate (lr), alpha ($\alpha$), and rank ($r$).

| Method | Name | CB | RTE | BoolQ | WiC | WSC | CoLA | SST-2 | MRPC | Amazon | PubMed |
|---|---|---|---|---|---|---|---|---|---|---|---|
| FT | bs | 16 | 16 | 32 | 32 | 16 | 32 | 32 | 16 | 16 | 8 |
| | lr | $5e-5$ | $1e-5$ | $1e-5$ | $1e-5$ | $1e-3$ | $5e-5$ | $1e-5$ | $1e-5$ | $1e-5$ | $5e-6$ |
| LoRA | bs | 16 | 16 | 32 | 16 | 16 | 16 | 32 | 32 | 16 | 8 |
| | lr | $1e-3$ | $1e-3$ | $5e-4$ | $1e-3$ | $1e-4$ | $1e-3$ | $5e-4$ | $5e-4$ | $1e-3$ | $5e-4$ |
| | $\alpha$ | | | | | | 8 | | | | |
| | $r$ | | | | | | 8 | | | | |
| LP-FT | bs | | | | | | 8 | | | | |
| | lr | $5e-6$ | $1e-5$ | $1e-5$ | $1e-5$ | $1e-3$ | $1e-5$ | $1e-5$ | $1e-5$ | $1e-6$ | $5e-6$ |
| LP-LoRA | bs | | | | | | 8 | | | | |
| | lr | $1e-4$ | $5e-4$ | $5e-4$ | $1e-3$ | $1e-4$ | $1e-3$ | $1e-3$ | $1e-3$ | $5e-4$ | $1e-3$ |
| | $\alpha$ | | | | | | 8 | | | | |
| | $r$ | | | | | | 8 | | | | |

Table 7: Dataset statistics. This table provides detailed counts of the classes, training, validation, and test samples for different datasets across various tasks including natural language inference (NLI), word sense disambiguation (WSD), question answering (QA), coreference resolution (coref.), sentiment analysis (sentiment), and sequential sentence classification (sequential).

| Dataset | Benchmark | Classes | Train | Val | Test | Task |
|---|---|---|---|---|---|---|
| CB | | 3 | 225 | 25 | 57 | NLI |
| RTE | | 2 | 2,241 | 249 | 277 | NLI |
| BoolQ | SuperGLUE | 2 | 8,484 | 943 | 3,270 | QA |
| WiC | | 2 | 5,400 | 600 | 638 | WSD |
| WSC | | 2 | 498 | 56 | 104 | coref. |
| CoLA | | 2 | 7,695 | 855 | 1,040 | acceptability |
| SST-2 | GLUE | 2 | 60,614 | 6,735 | 872 | sentiment |
| MRPC | | 2 | 3,301 | 367 | 408 | sentiment |
| Amazon | | 3 | 27,000 | 3,000 | 38,905 | sentiment |
| Dynasent | | 3 | - | - | 4,320 | sentiment |
| SemEval | BOSS | 3 | - | - | 20,622 | sentiment |
| SST-5 | | 3 | - | - | 1,067 | sentiment |
| PubMed 20k RCT | PubMed | 5 | 15,000 | 2,500 | 2,500 | sequential |

Table 8: Test results on the SuperGLUE and GLUE benchmarks. We report the accuracy and its standard deviation, other than the CoLA dataset, which is evaluated by the Matthew's correlation coefficient. We take the average of five seeds.

| Dataset | LP | FT | LP-FT | LoRA | LP-LoRA |
|---|---|---|---|---|---|
| CB | $77.86 \pm 4.24$ | $81.43 \pm 3.91$ | $\mathbf{84.64 \pm 2.40}$ | $77.50 \pm 5.30$ | $75.71 \pm 2.04$ |
| RTE | $57.69 \pm 1.10$ | $74.73 \pm 3.04$ | $\mathbf{76.75 \pm 0.87}$ | $72.85 \pm 1.41$ | $74.08 \pm 2.57$ |
| SST-2 | $86.31 \pm 0.10$ | $92.41 \pm 0.32$ | $\mathbf{94.52 \pm 0.26}$ | $50.92 \pm 0.00$ | $94.22 \pm 0.45$ |
| WIC | $61.32 \pm 0.28$ | $65.89 \pm 1.15$ | $\mathbf{66.14 \pm 1.83}$ | $62.70 \pm 7.37$ | $64.29 \pm 1.82$ |
| CoLA | $46.27 \pm 0.33$ | $\mathbf{58.75 \pm 1.70}$ | $57.95 \pm 1.95$ | $57.29 \pm 2.98$ | $58.21 \pm 1.55$ |
| MRPC | $73.09 \pm 0.86$ | $\mathbf{88.14 \pm 0.73}$ | $87.60 \pm 0.79$ | $68.38 \pm 0.00$ | $87.79 \pm 1.00$ |
| WSC | $\mathbf{63.46 \pm 0.00}$ | $\mathbf{63.46 \pm 0.00}$ | $\mathbf{63.46 \pm 0.00}$ | $63.46 \pm 0.68$ | $\mathbf{63.46 \pm 0.00}$ |
| BoolQ | $64.66 \pm 0.08$ | $78.69 \pm 0.27$ | $\mathbf{79.00 \pm 0.42}$ | $77.59 \pm 0.39$ | $77.67 \pm 0.50$ |

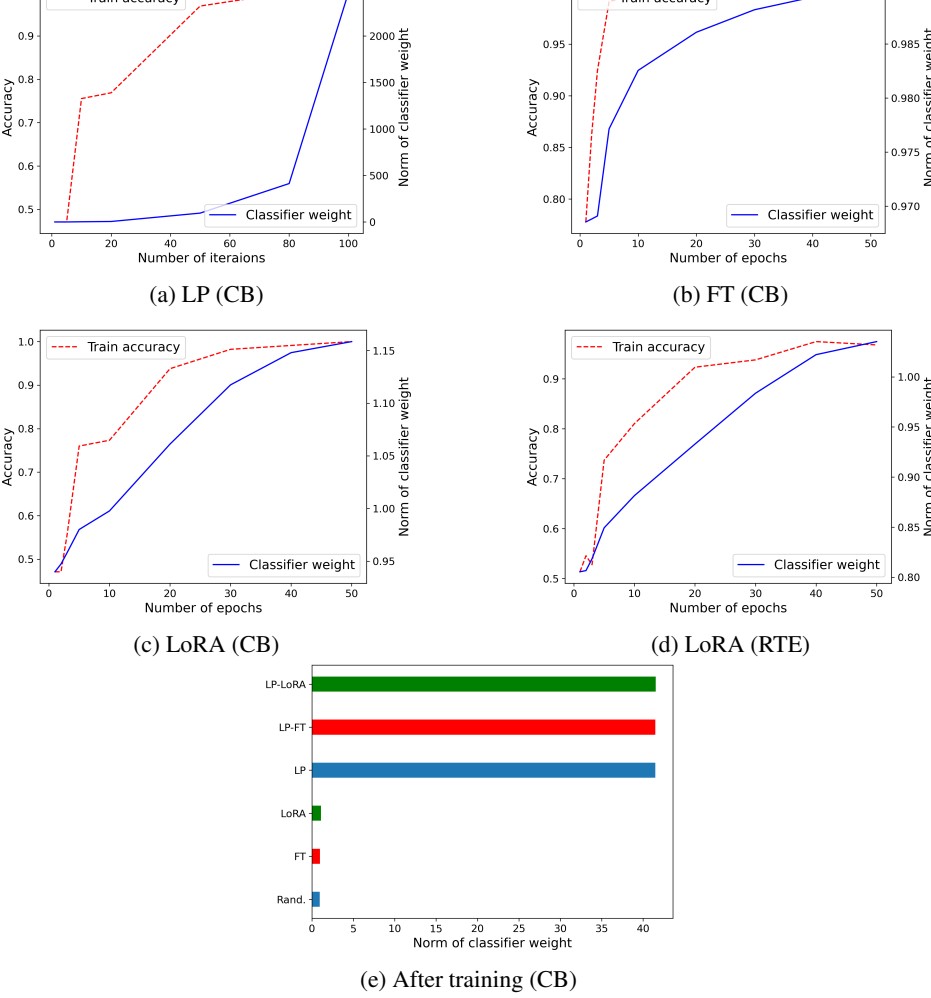

(a) LP (CB)

(b) FT (CB)

(c) LoRA (CB)

(d) LoRA (RTE)

(e) After training (CB)

Figure 4: The increase in the norm of the classifier weight during training.

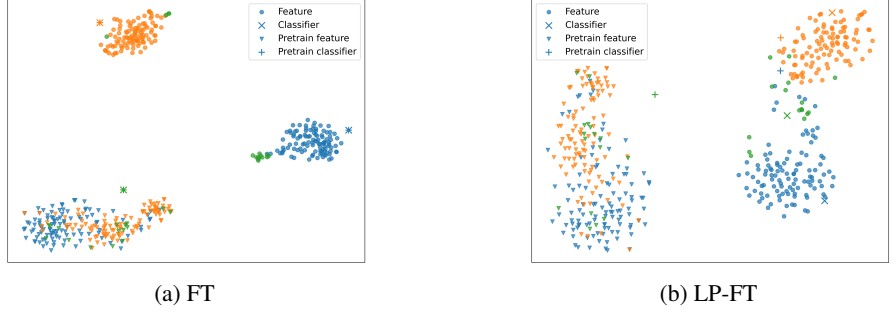

(a) FT

(b) LP-FT

Figure 5: Small changes in feature and large changes in classifier weight during LP-FT. We visualize the t-SNE plot of the penultimate layer features and the classifier row vector of the model trained on the CB dataset. (a) The features after FT are clearly separated by class, while the classifier row vectors are plotted nearly the same place as the pre-trained model. (b) The features after LP-FT keep the structure of the pre-trained model, while the classifier row vectors are changed from the initialization.

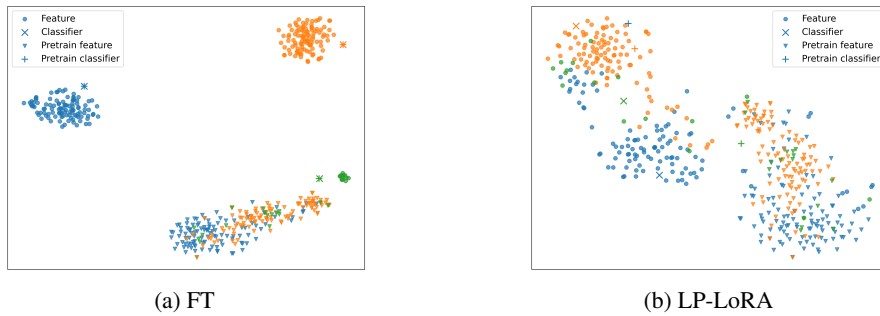

(a) FT                                      (b) LP-LoRA

Figure 6: The t-SNE plot of the penultimate layer features and the classifier row vector of the model trained with LoRA on the CB dataset.

### A.4.2 Results of NTK analysis

Table 9 displays the kernel statistics, while Figure 7 shows the distribution of singular values. Figure 8 and Figure 9 visually depict the trace norms of sub-matrices within the NTK matrix. For the kernel matrix $K \in \mathbb{R}^{NC \times NC}$, we calculated the trace norms of the sub-matrix $K(x_i, x_j) \in \mathbb{R}^{C \times C}$ for each sample pair $(x_i, x_j)$ in the training sets.

Figure 8 reveals a consistent pattern in the FT-effective component of the NTK matrix across all datasets: pairs of identical samples in diagonal positions typically exhibit higher trace norms. This suggests that the FT-effective component is more effective at capturing relationships among samples compared to the pre-train-effective component. Additionally, in the CB dataset, certain sample pairs, particularly in classes 1 and 3, show notably high trace norms, indicating that the pre-trained model effectively differentiates between these class samples.

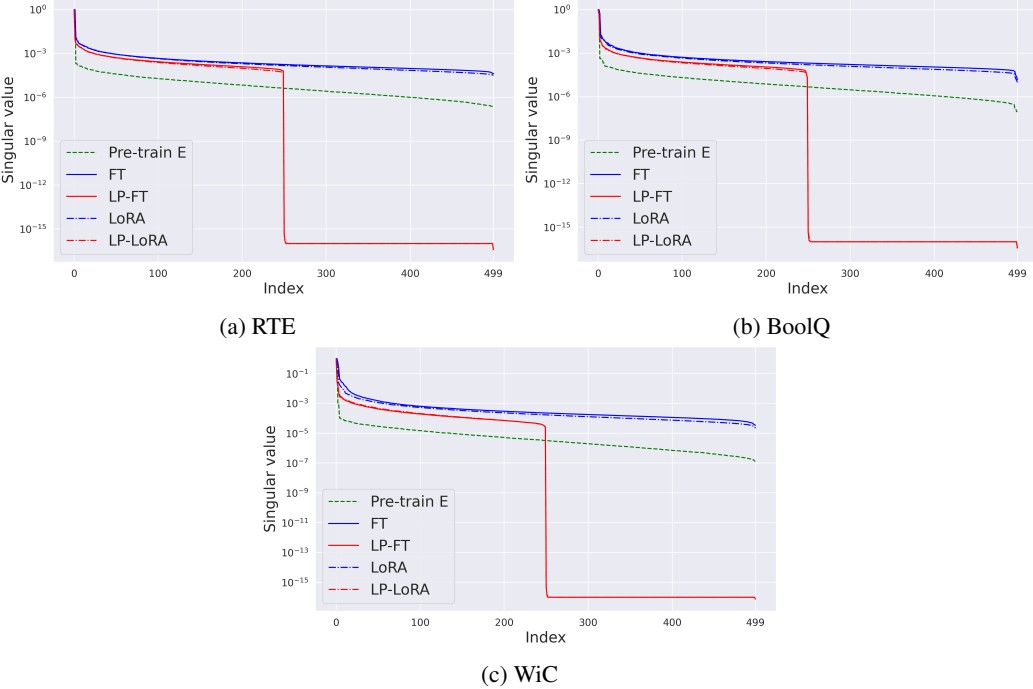

(a) RTE                                      (b) BoolQ

(c) WiC

Figure 7: Singular value distribution normalized by the maximum singular value on the RTE, BoolQ, and WiC datasets. Pre-train E denotes the pre-train-effective component, and other plots denote the FT-effective component of NTK matrix with each training option.

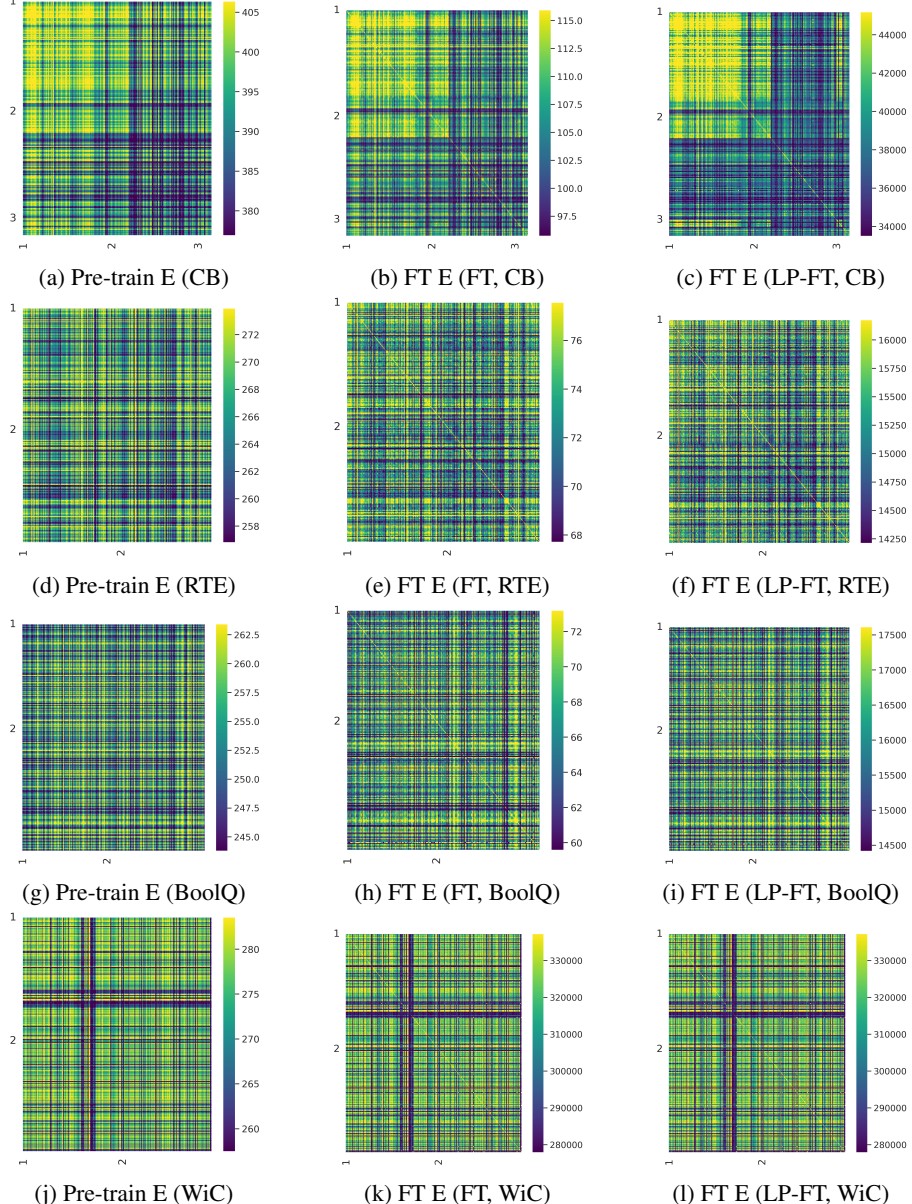

Figure 8: Heat map of NTK matrix on the CB, RTE, BoolQ, and WiC dataset. We calculate the trace norm of the sub-matrix of the NTK matrix for each sample pair and visualize them grouped by class. Pre-train E and FT E refer to the pre-train-effective and FT-effective components of the NTK matrix.

Table 9: Kernel statistics on the RTE, BoolQ, and WiC datasets. FN, Acc, and FT Ratio denote the Frobenius norm, kernel regression accuracy, and contribution of the FT-effective component, respectively. Pre-train E and FT E refer to the pre-train-effective and FT-effective components of the NTK matrix.

| Dataset | Method | Kernel | Rank | FN | Acc (train/test) | FT Ratio |
|---|---|---|---|---|---|---|
| RTE | - | Pre-train E | 28 | $4.70 \times 10^4$ | 66.40/51.20 | - |
| | FT | FT E | 488 | $1.29 \times 10^4$ | 96.60/53.40 | 0.2148 |
| | | NTK | 191 | $5.98 \times 10^4$ | 97.60/53.00 | |
| | LoRA | FT E | 432 | $2.51 \times 10^1$ | 70.80/54.60 | 0.0005 |
| | | NTK | 30 | $4.70 \times 10^4$ | 59.60/54.80 | |
| | LP-FT | FT E | 250 | $3.80 \times 10^6$ | 100.00/51.20 | 0.9918 |
| | | NTK | 251 | $3.84 \times 10^6$ | 100.00/51.20 | |
| | LP-LoRA | FT E | 243 | $7.60 \times 10^3$ | 84.80/51.20 | 0.1942 |
| | | NTK | 103 | $5.26 \times 10^4$ | 88.00/51.20 | |
| BoolQ | - | Pre-train E | 32 | $4.48 \times 10^4$ | 53.60/57.20 | - |
| | FT | FT E | 495 | $1.24 \times 10^4$ | 100.00/56.40 | 0.2139 |
| | | NTK | 215 | $5.67 \times 10^4$ | 53.80/57.20 | |
| | LoRA | FT E | 448 | $2.48 \times 10^1$ | 53.60/57.20 | 0.0005 |
| | | NTK | 34 | $4.48 \times 10^4$ | 53.60/57.20 | |
| | LP-FT | FT E | 247 | $4.46 \times 10^6$ | 100.00/61.60 | 0.9921 |
| | | NTK | 248 | $4.49 \times 10^6$ | 100.00/61.20 | |
| | LP-LoRA | FT E | 237 | $8.56 \times 10^3$ | 68.80/63.60 | 0.2118 |
| | | NTK | 99 | $5.07 \times 10^4$ | 86.00/59.20 | |
| WiC | - | Pre-train E | 16 | $4.81 \times 10^4$ | 66.00/54.00 | - |
| | FT | FT E | 488 | $1.45 \times 10^4$ | 89.00/59.00 | 0.2216 |
| | | NTK | 235 | $6.17 \times 10^4$ | 90.60/59.00 | - |
| | LoRA | FT E | 438 | $2.58 \times 10^1$ | 72.00/52.00 | 0.0005 |
| | | NTK | 19 | $4.81 \times 10^4$ | 65.80/56.40 | |
| | LP-FT | FT E | 218 | $7.77 \times 10^7$ | 100.00/56.80 | 0.9996 |
| | | NTK | 219 | $7.77 \times 10^7$ | 100.00/56.40 | |
| | LP-LoRA | FT E | 218 | $1.09 \times 10^5$ | 72.00/59.60 | 0.7454 |
| | | NTK | 195 | $1.47 \times 10^5$ | 80.80/59.60 | |

### A.4.3   Experiments on BOSS benchmark

Table 10 shows indicate that LP-FT surpasses FT in OOD robustness and achieves higher accuracy in ID settings than LoRA. This suggests that LP-FT is effective in enhancing model robustness to OOD samples with reduced feature changes.

Table 11 displays the statistics of feature and classifier changes on the Amazon, Dynasent, SemEval, and SST-5 datasets. The FDR within the ID is lower for LP-FT than for FT, whereas the FDR for OOD is higher for LP-FT than for FT. This indicates that LP-FT is learning robust features that are less sensitive to OOD data.

### A.4.4   Change of feature and classifier norms

Table 12 shows the changes in features during the FT stage, indicating that the changes are smaller during LP-FT compared to FT. Table 13 shows the classifier norms, which increase during training, with a more noticeable increase observed during LP than during FT.

Table 10: Evaluation results on BOSS benchmark. We report the average accuracy and standard deviation over five seeds. The best results are highlighted in bold.

| Method | ID | OOD | | |
| --- | --- | --- | --- | --- |
| | Amazon | Dynasent | SemEval | SST-5 |
| LP | $83.04 \pm 0.01$ | $42.69 \pm 0.05$ | $50.04 \pm 0.01$ | $56.81 \pm 0.11$ |
| FT | $88.66 \pm 1.62$ | $44.33 \pm 1.11$ | $52.20 \pm 1.82$ | $72.52 \pm 1.28$ |
| LoRA | $86.05 \pm 2.16$ | $\mathbf{46.70 \pm 1.68}$ | $\mathbf{55.29 \pm 2.93}$ | $72.88 \pm 1.84$ |
| LP-FT | $\mathbf{88.89 \pm 1.02}$ | $45.41 \pm 0.80$ | $51.96 \pm 2.72$ | $\mathbf{73.78 \pm 1.05}$ |
| LP-LoRA | $88.17 \pm 1.97$ | $43.37 \pm 1.50$ | $48.84 \pm 3.20$ | $72.31 \pm 1.30$ |

Table 11: Comparison of feature and classifier changes on the Amazon (ID), Dynasent, SemEval, and SST-5 (OOD) datasets. CS, Diff, FDR, and Norm denote cosine similarity, difference norm, Fisher's discriminant ratio, and norm, respectively. (F) and (C) indicate feature and classifier statistics. Averages were calculated over five seeds.

| Method | Amazon | | | | Dynasent | | |
| --- | --- | --- | --- | --- | --- | --- | --- |
| | CS(F) | Diff(F) | FDR(F) | Norm(C) | CS(F) | Diff(F) | FDR(F) |
| Pre-trained | 0.996 | – | $1.30 \times 10^0$ | $9.51 \times 10^{-1}$ | 0.996 | – | $1.94 \times 10^0$ |
| LP | 0.996 | – | $1.30 \times 10^0$ | $1.20 \times 10^2$ | 0.996 | – | $1.94 \times 10^0$ |
| FT | 0.691 | $1.94 \times 10^1$ | $3.74 \times 10^0$ | $9.50 \times 10^{-1}$ | 0.652 | $1.80 \times 10^1$ | $2.03 \times 10^0$ |
| LoRA | 0.848 | $1.16 \times 10^1$ | $3.38 \times 10^0$ | $1.81 \times 10^0$ | 0.855 | $7.53 \times 10^0$ | $2.06 \times 10^0$ |
| LP-FT | 0.999 | $2.27 \times 10^0$ | $3.00 \times 10^0$ | $1.20 \times 10^2$ | 0.998 | $2.54 \times 10^0$ | $2.20 \times 10^0$ |
| LP-LoRA | 0.999 | $2.24 \times 10^0$ | $3.01 \times 10^0$ | $1.18 \times 10^2$ | 0.999 | $2.56 \times 10^0$ | $2.04 \times 10^0$ |

| Method | SemEval | | | SST5 | | |
| --- | --- | --- | --- | --- | --- | --- |
| | CS(F) | Diff(F) | FDR(F) | CS(F) | Diff(F) | FDR(F) |
| Pre-trained | 0.996 | – | $1.24 \times 10^0$ | 0.998 | – | $1.69 \times 10^1$ |
| LP | 0.996 | – | $1.24 \times 10^0$ | 0.998 | – | $1.69 \times 10^1$ |
| FT | 0.727 | $1.68 \times 10^1$ | $1.49 \times 10^0$ | 0.604 | $1.84 \times 10^1$ | $2.26 \times 10^1$ |
| LoRA | 0.885 | $6.74 \times 10^0$ | $1.44 \times 10^0$ | 0.837 | $8.72 \times 10^0$ | $2.01 \times 10^1$ |
| LP-FT | 0.997 | $2.06 \times 10^0$ | $1.45 \times 10^0$ | 0.998 | $1.86 \times 10^0$ | $2.02 \times 10^1$ |
| LP-LoRA | 0.999 | $2.08 \times 10^0$ | $1.19 \times 10^0$ | 0.998 | $1.85 \times 10^0$ | $1.95 \times 10^1$ |

Table 12: Feature change in FT stage. The change during LP-FT is smaller than during FT.

| Dataset | FT | LoRA | LP-FT | LP-LoRA |
| --- | --- | --- | --- | --- |
| CB | $2.11 \times 10^1$ | $2.07 \times 10^1$ | $1.15 \times 10^1$ | $7.85 \times 10^0$ |
| RTE | $2.12 \times 10^1$ | $1.51 \times 10^1$ | $3.33 \times 10^0$ | $3.87 \times 10^0$ |
| COLA | $1.91 \times 10^1$ | $1.10 \times 10^1$ | $3.05 \times 10^0$ | $2.75 \times 10^0$ |
| SST-2 | $2.31 \times 10^1$ | $3.78 \times 10^0$ | $6.95 \times 10^0$ | $2.17 \times 10^0$ |
| MRPC | $2.11 \times 10^1$ | $1.80 \times 10^0$ | $1.84 \times 10^0$ | $1.94 \times 10^0$ |
| BoolQ | $2.23 \times 10^1$ | $1.55 \times 10^1$ | $2.31 \times 10^0$ | $1.95 \times 10^0$ |
| WiC | $2.08 \times 10^1$ | $1.04 \times 10^1$ | $2.28 \times 10^0$ | $2.16 \times 10^0$ |
| WSC | $9.14 \times 10^0$ | $2.44 \times 10^{-1}$ | $7.33 \times 10^0$ | $2.02 \times 10^{-1}$ |
| Amazon | $1.98 \times 10^1$ | $1.35 \times 10^1$ | $2.21 \times 10^0$ | $2.28 \times 10^0$ |
| Dynasent | $1.94 \times 10^1$ | $8.02 \times 10^0$ | $2.47 \times 10^0$ | $2.59 \times 10^0$ |
| SemEval | $1.83 \times 10^1$ | $6.97 \times 10^0$ | $1.99 \times 10^0$ | $2.14 \times 10^0$ |
| SST-5 | $2.03 \times 10^1$ | $9.08 \times 10^0$ | $1.79 \times 10^0$ | $1.89 \times 10^0$ |

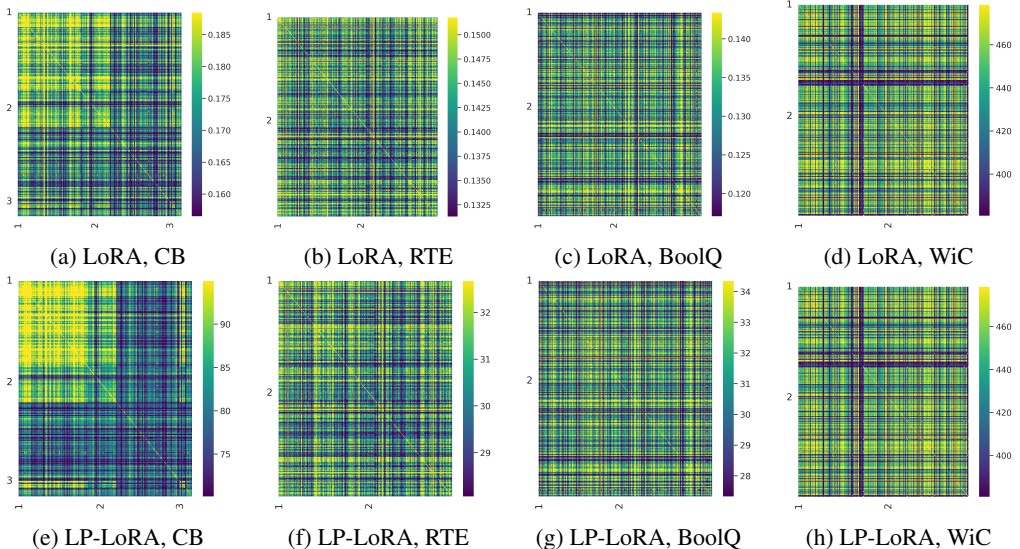

| (a) LoRA, CB | (b) LoRA, RTE | (c) LoRA, BoolQ | (d) LoRA, WiC |

| (e) LP-LoRA, CB | (f) LP-LoRA, RTE | (g) LP-LoRA, BoolQ | (h) LP-LoRA, WiC |

Figure 9: Heat map of NTK matrix of FT-effective component with LoRA on the CB, RTE, BoolQ, and WiC dataset with LoRA. We calculate the trace norm of the sub-matrix of the NTK matrix for each sample pair and visualize them grouped by class.

Table 13: The classifier weight norms. The classifier weight norms increase during training, and the increase is more pronounced in LP.

| Dataset | Pretrain | FT | LoRA | LP | LP-FT | LP-LoRA |
|---|---|---|---|---|---|---|
| CB | $9.47 \times 10^{-1}$ | $9.51 \times 10^{-1}$ | $1.56 \times 10^{0}$ | $3.35 \times 10^{1}$ | $3.35 \times 10^{1}$ | $3.35 \times 10^{1}$ |
| RTE | $7.95 \times 10^{-1}$ | $8.05 \times 10^{-1}$ | $1.45 \times 10^{0}$ | $2.86 \times 10^{1}$ | $2.86 \times 10^{1}$ | $2.85 \times 10^{1}$ |
| COLA | $7.95 \times 10^{-1}$ | $7.88 \times 10^{-1}$ | $1.06 \times 10^{0}$ | $3.46 \times 10^{1}$ | $3.46 \times 10^{1}$ | $3.51 \times 10^{1}$ |
| SST2 | $7.95 \times 10^{-1}$ | $7.20 \times 10^{-1}$ | $1.96 \times 10^{0}$ | $1.32 \times 10^{2}$ | $1.09 \times 10^{2}$ | $1.03 \times 10^{2}$ |
| MRPC | $7.95 \times 10^{-1}$ | $7.98 \times 10^{-1}$ | $1.35 \times 10^{0}$ | $1.12 \times 10^{1}$ | $1.12 \times 10^{1}$ | $1.12 \times 10^{1}$ |
| BoolQ | $7.95 \times 10^{-1}$ | $7.98 \times 10^{-1}$ | $1.15 \times 10^{0}$ | $1.27 \times 10^{1}$ | $1.27 \times 10^{1}$ | $1.25 \times 10^{1}$ |
| WiC | $7.95 \times 10^{-1}$ | $7.98 \times 10^{-1}$ | $1.14 \times 10^{0}$ | $3.21 \times 10^{1}$ | $3.25 \times 10^{1}$ | $3.27 \times 10^{1}$ |
| WSC | $7.95 \times 10^{-1}$ | $6.87 \times 10^{-1}$ | $7.88 \times 10^{-1}$ | $2.26 \times 10^{-4}$ | $1.08 \times 10^{-1}$ | $2.16 \times 10^{-2}$ |
| Amazon | $9.51 \times 10^{-1}$ | $9.47 \times 10^{-1}$ | $1.67 \times 10^{0}$ | $1.21 \times 10^{2}$ | $1.21 \times 10^{2}$ | $1.20 \times 10^{2}$ |

### A.4.5   Effects of classifier weight norms in training

Figure 10 (Boss benchmark) and Figure 11 (the CB and RTE datasets) illustrate the changes in features from the pre-trained models. Except for the CB dataset, the change in features in LP-FT is generally smaller than in FT when using large classifier norms. The CB dataset has a smaller sample size, which could be an exception.

### A.4.6   Temperature scaling

The result of temperature scaling on SuperGLUE and GLUE is presented in Tables 14 and 15.

### A.4.7   PubMed 20k

In addition to the natural language understanding benchmarks, we also evaluated LP-FT on the PubMed 20k RCT dataset to evaluate its effectiveness in practical applications. The PubMed 20k RCT dataset, a subset of PubMed 200k [Dernoncourt and Lee, 2017], comprises 20,000 medical abstracts from randomized controlled trials, categorized into five classes. Efficient tools for navigating extensive medical literature are essential for the medical community.

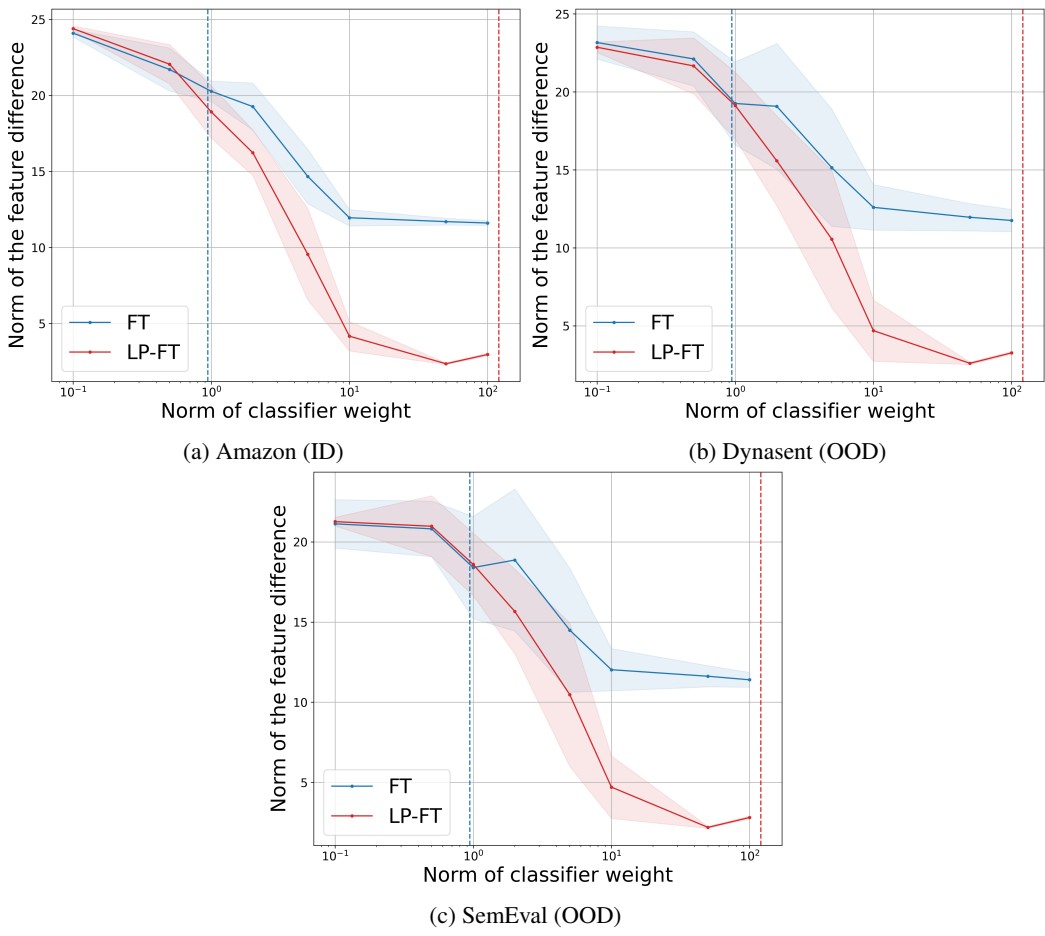

Figure 10: Difference of features of the samples with scaling the classifier weight norms on BOSS benchmark. Solid lines show mean values; shaded areas represent standard errors. The dashed vertical lines indicate the original norms of the classifier weight.

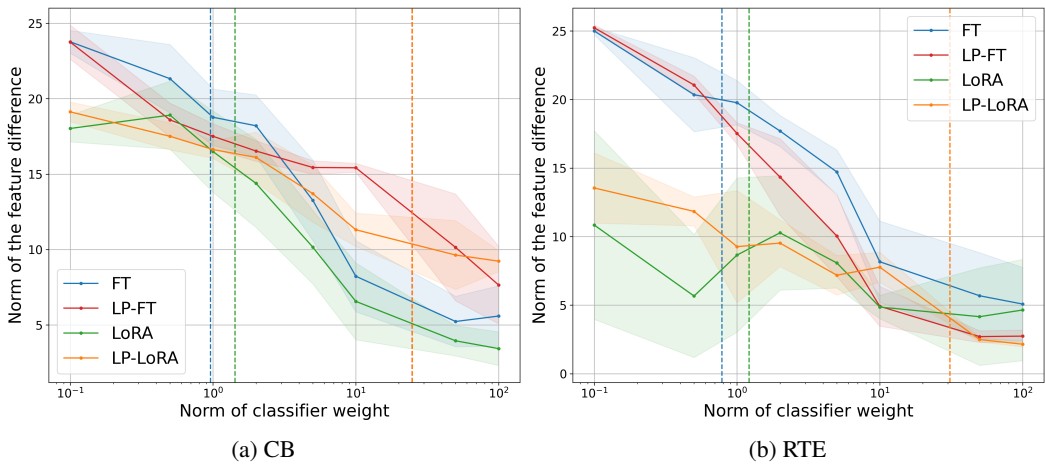

Figure 11: Difference of features of the samples with scaling the classifier weight norms on the CB and RTE datasets. Solid lines show mean values; shaded areas represent standard errors. The dashed vertical lines indicate the original norms of the classifier weight.

Table 14: ECE and MCE with temperature scaling on SuperGLUE. w/o TS and w/ TS denote without and with temperature scaling, respectively, and Imp. represents the improvement because of temperature scaling. We bold the best improvements. We take 5 seeds and report the mean and the standard deviation.

| Dataset | Metric | Method | w/o TS | w/ TS | Imp. |
|---|---|---|---|---|---|
| CB | ECE (%) | FT | $15.60 \pm 0.96$ | $14.64 \pm 1.75$ | 0.95 |
| | | LP-FT | $13.93 \pm 0.45$ | $13.13 \pm 0.56$ | 0.80 |
| | | LoRA | $12.89 \pm 0.41$ | $16.22 \pm 0.55$ | $-3.34$ |
| | | LP-LoRA | $14.78 \pm 0.93$ | $13.51 \pm 1.67$ | 1.27 |
| | MCE (%) | FT | $75.99 \pm 6.12$ | $69.99 \pm 5.83$ | 6.01 |
| | | LP-FT | $76.78 \pm 3.66$ | $70.28 \pm 3.27$ | 6.50 |
| | | LoRA | $52.58 \pm 4.72$ | $66.75 \pm 7.96$ | $-14.16$ |
| | | LP-LoRA | $68.16 \pm 4.95$ | $60.80 \pm 2.30$ | 7.36 |
| RTE | ECE (%) | FT | $21.16 \pm 1.36$ | $5.13 \pm 0.63$ | 16.03 |
| | | LP-FT | $21.72 \pm 0.28$ | $5.48 \pm 0.77$ | 16.24 |
| | | LoRA | $11.92 \pm 2.23$ | $6.17 \pm 0.20$ | 5.76 |
| | | LP-LoRA | $18.14 \pm 0.99$ | $5.72 \pm 0.48$ | 12.42 |
| | MCE (%) | FT | $53.11 \pm 8.51$ | $25.87 \pm 6.30$ | 27.24 |
| | | LP-FT | $63.95 \pm 7.70$ | $13.94 \pm 1.80$ | 50.01 |
| | | LoRA | $25.04 \pm 3.33$ | $13.75 \pm 0.91$ | 11.29 |
| | | LP-LoRA | $40.46 \pm 7.22$ | $18.82 \pm 2.00$ | 21.63 |
| BoolQ | ECE (%) | FT | $13.63 \pm 0.61$ | $1.83 \pm 0.09$ | 11.81 |
| | | LP-FT | $18.93 \pm 0.15$ | $2.41 \pm 0.42$ | 16.51 |
| | | LoRA | $8.88 \pm 0.38$ | $1.45 \pm 0.18$ | 7.43 |
| | | LP-LoRA | $14.09 \pm 0.92$ | $2.07 \pm 0.19$ | 12.02 |
| | MCE (%) | FT | $23.26 \pm 1.48$ | $5.79 \pm 0.90$ | 17.47 |
| | | LP-FT | $40.82 \pm 1.94$ | $5.21 \pm 0.53$ | 35.60 |
| | | LoRA | $13.96 \pm 0.72$ | $3.85 \pm 0.56$ | 10.11 |
| | | LP-LoRA | $24.60 \pm 2.52$ | $5.51 \pm 0.72$ | 19.09 |
| WiC | ECE (%) | FT | $25.88 \pm 2.39$ | $8.85 \pm 0.53$ | 17.03 |
| | | LP-FT | $29.47 \pm 1.57$ | $7.68 \pm 0.55$ | 21.78 |
| | | LoRA | $18.66 \pm 4.39$ | $5.93 \pm 1.42$ | 12.73 |
| | | LP-LoRA | $22.22 \pm 1.98$ | $8.06 \pm 0.60$ | 14.15 |
| | MCE (%) | FT | $41.59 \pm 5.39$ | $17.01 \pm 2.87$ | 24.58 |
| | | LP-FT | $39.20 \pm 2.74$ | $17.04 \pm 1.50$ | 22.16 |
| | | LoRA | $27.95 \pm 7.38$ | $11.40 \pm 2.77$ | 16.54 |
| | | LP-LoRA | $30.99 \pm 3.64$ | $14.45 \pm 1.01$ | 16.54 |
| WSC | ECE (%) | FT | $6.26 \pm 2.37$ | $7.97 \pm 0.06$ | $-1.71$ |
| | | LP-FT | $6.38 \pm 1.78$ | $8.01 \pm 0.06$ | $-1.63$ |
| | | LoRA | $10.53 \pm 1.35$ | $9.19 \pm 0.60$ | 1.34 |
| | | LP-LoRA | $11.40 \pm 0.23$ | $8.24 \pm 0.01$ | 3.15 |
| | MCE (%) | FT | $6.26 \pm 2.37$ | $7.97 \pm 0.06$ | $-1.71$ |
| | | LP-FT | $6.38 \pm 1.78$ | $8.01 \pm 0.06$ | $-1.63$ |
| | | LoRA | $13.27 \pm 1.12$ | $11.12 \pm 1.51$ | 2.15 |
| | | LP-LoRA | $11.40 \pm 0.23$ | $8.24 \pm 0.01$ | 3.15 |

The results are presented in Table 16. The LoRA model outperforms other models, although the performance of FT, LP-FT, and LoRA models are relatively similar.

Table 15: ECE and MCE with temperature scaling on GLUE. w/o TS and w/ TS denote without and with temperature scaling, respectively, and Imp. represents the improvement because of temperature scaling. We bold the best improvements. We take 5 seeds and report the mean and the standard deviation.

| Dataset | Metric | Method | w/o TS | w/ TS | Imp. |
|---------|--------|--------|--------|-------|------|
| CoLA | ECE (%) | FT | $15.08 \pm 0.55$ | $4.46 \pm 0.83$ | 10.61 |
| | | LP-FT | $15.74 \pm 0.40$ | $9.53 \pm 1.23$ | 6.21 |
| | | LoRA | $11.25 \pm 1.32$ | $4.18 \pm 0.40$ | 7.07 |
| | | LP-LoRA | $13.82 \pm 0.48$ | $4.30 \pm 0.43$ | 9.52 |
| | MCE (%) | FT | $47.19 \pm 5.15$ | $24.35 \pm 3.33$ | 22.84 |
| | | LP-FT | $54.59 \pm 2.94$ | $20.31 \pm 1.37$ | 34.28 |
| | | LoRA | $31.01 \pm 5.83$ | $15.23 \pm 2.74$ | 15.78 |
| | | LP-LoRA | $38.36 \pm 7.85$ | $15.36 \pm 1.83$ | 23.00 |
| SST-2 | ECE (%) | FT | $4.61 \pm 0.31$ | $2.26 \pm 0.22$ | 2.35 |
| | | LP-FT | $5.67 \pm 0.12$ | $2.00 \pm 0.21$ | 3.66 |
| | | LoRA | $4.84 \pm 0.13$ | $2.71 \pm 0.16$ | 2.12 |
| | | LP-LoRA | $6.22 \pm 0.10$ | $2.53 \pm 0.08$ | 3.69 |
| | MCE (%) | FT | $49.22 \pm 4.78$ | $42.72 \pm 5.24$ | 6.50 |
| | | LP-FT | $74.91 \pm 1.72$ | $42.77 \pm 5.75$ | 32.13 |
| | | LoRA | $54.20 \pm 2.84$ | $36.58 \pm 5.82$ | 17.63 |
| | | LP-LoRA | $71.12 \pm 3.97$ | $32.47 \pm 3.74$ | 38.65 |
| MRPC | ECE (%) | FT | $10.71 \pm 0.39$ | $4.61 \pm 0.24$ | 6.10 |
| | | LP-FT | $10.35 \pm 0.14$ | $3.68 \pm 0.10$ | 6.68 |
| | | LoRA | $6.58 \pm 0.68$ | $4.04 \pm 0.87$ | 2.54 |
| | | LP-LoRA | $9.03 \pm 0.85$ | $3.89 \pm 0.40$ | 5.14 |
| | MCE (%) | FT | $61.84 \pm 7.93$ | $32.72 \pm 1.69$ | 29.12 |
| | | LP-FT | $74.43 \pm 2.22$ | $22.73 \pm 1.33$ | 51.70 |
| | | LoRA | $28.80 \pm 5.05$ | $17.57 \pm 2.00$ | 11.23 |
| | | LP-LoRA | $52.20 \pm 6.64$ | $22.76 \pm 7.60$ | 29.44 |

Table 16: Test accuracy on PubMed 20k.

| LP | FT | LP-FT | LoRA | LP-LoRA |
|----|----|-------|------|---------|
| $82.64 \pm 0.02$ | $87.09 \pm 0.17$ | $87.05 \pm 0.11$ | $\mathbf{87.13 \pm 0.09}$ | $86.85 \pm 0.07$ |

