# OpenReview forum: "Understanding Linear Probing then Fine-tuning Language Models from NTK Perspective"
_NeurIPS.cc/2024/Conference — NeurIPS 2024 poster_

### Official Review · Reviewer_qfrt · 2024-07-13

**Soundness:** 3
**Presentation:** 3
**Contribution:** 3
**Rating:** 6
**Confidence:** 2

**Summary:**

The authors analyze the training dynamics of LP-FT for classification models based on NTK theory. They decomposed NTK matrix and highlight the importance of linear head norm. Additionally, they highlight the increased linear head norms can negatively affect the model calibration and can be fixed by temperature scaling. Finally, they extend their analysis to the LoRA method.

**Strengths:**

1. The application of NTK theory to analyze LP-FT in complex models like Transformers is interesting.
2. The theoretical analysis is thorough, well-supported by mathematical proofs, and aligns with empirical observations.

**Weaknesses:**

1. The experiments primarily focus on the classification tasks. Since the authors also extend their analysis to LoRA, exploring additional domains and harder tasks (reasoning, math, code generation) combined with LLMs could siginificantly strengthen the generalizability of the findings.

**Questions:**

N/A

---

> ### Author Rebuttal · Authors · 2024-08-04
>
> Thank you very much for taking the time and effort to review our paper. We appreciate your insightful question.
> > Q1: The experiments primarily focus on the classification tasks. Since the authors also extend their analysis to LoRA, exploring additional domains and harder tasks (reasoning, math, code generation) combined with LLMs could significantly strengthen the generalizability of the findings.
>
> A: We conducted experiments on question-answering tasks with the RoBERTa model on the SQuAD1.1 [1] and SQuAD2.0 [2] datasets. The following results show the following:
> 1. Smaller feature changes in LP-FT.
> 1. A significant increase in classifier weight norms during LP.
> 1. Effectiveness of temperature scaling on LP-FT.
>
> These points validate our analysis. Although the F1 score of LP-FT on SQuAD1.1 is lower than that of standard fine-tuning (FT), this result is influenced by the choice of hyperparameters and could improve with further optimization.
>
> ### F1 score on test set
> | Dataset | LP             | FT             | LoRA           | LP-FT          | LP-LoRA        |
> |---------|----------------|----------------|----------------|----------------|----------------|
> | SQuAD1.1   | 24.14 ± 0.15   | 91.72 ± 0.05   | 91.57 ± 0.07   | **91.80 ± 0.09**   | 91.64 ± 0.06   |
> | SQuAD2.0   | 22.69 ± 0.21   | **82.50 ± 0.44**   | 80.74 ± 1.56   | 81.31 ± 0.50   | 80.72 ± 0.22   |
>
> ### Classifier weight norms
> | Dataset | Pretrain     | FT          | LoRA        | LP          | LP-FT       | LP-LoRA     |
> |---------|--------------|-------------|-------------|-------------|-------------|-------------|
> | SQuAD1.1   | 7.82e-01     | 8.50e-01    | 8.72e-01    | 1.21e+01    | 1.21e+01    | 1.21e+01    |
> | SQuAD2.0   | 7.82e-01     | 8.41e-01    | 8.71e-01    | 1.26e+01    | 1.26e+01    | 1.26e+01    |
>
> ### Norm of the feature difference from the pre-trained model
> | Dataset | FT          | LoRA        | LP-FT       | LP-LoRA     |
> |---------|-------------|-------------|-------------|-------------|
> | SQuAD1.1   | 7.50e+00    | 7.49e+00    | 2.92e+00    | 2.94e+00    |
> | SQuAD2.0   | 7.59e+00    | 7.49e+00    | 3.17e+00    | 3.19e+00    |
>
> ### Effect of temperature scaling on the SQuAD1.1 dataset
> | Metric | Method   | w/o TS | w/ TS | Imp(%) |
> |--------|----------|--------|-------|--------|
> | ECE(%) | FT       | 15.94  | 8.09  | 49.25  |
> |        | LoRA     | 14.85  | 8.82  | 40.63  |
> |        | LP-FT    | 14.77  | 7.21  | **51.17**  |
> |        | LP-LoRA  | 14.97  | 7.33  | 51.05  |
> | MCE(%) | FT       | 25.64  | 14.06 | 45.17  |
> |        | LoRA     | 21.76  | 15.01 | 31.03  |
> |        | LP-FT    | 22.14  | 12.20 | 44.88  |
> |        | LP-LoRA  | 23.40  | 12.59 | **46.20**  |
>
> ### Effect of temperature scaling on the SQuAD2.0 dataset
>
> | Metric | Method   | w/o TS | w/ TS | Imp(%) |
> |--------|----------|--------|-------|--------|
> | ECE(%) | FT       | 7.42   | 1.67  | 77.50  |
> |        | LoRA     | 5.96   | 1.66  | 72.15  |
> |        | LP-FT    | 5.56   | 1.10  | **80.12**  |
> |        | LP-LoRA  | 4.68   | 1.24  | 73.46  |
> | MCE(%) | FT       | 13.66  | 5.23  | 61.73  |
> |        | LoRA     | 10.43  | 5.89  | 43.51  |
> |        | LP-FT    | 11.73  | 4.04  | **65.55**  |
> |        | LP-LoRA  | 9.22   | 5.68  | 38.43  |
>
> [1] Pranav Rajpurkar, Jian Zhang, Konstantin Lopyrev, and Percy Liang. Squad: 100,000+ questions for machine comprehension of text. arXiv preprint arXiv:1606.05250, 2016.
>
> [2] Pranav Rajpurkar, Robin Jia, and Percy Liang. Know what you don’t know: Unanswerable questions for squad. arXiv preprint arXiv:1806.03822, 2018.

---

> > ### Author Response · Authors · 2024-08-13
> >
> > Dear Reviewer qfrt,
> >
> > Thank you for your great efforts to review our paper.
> >
> > Does our response address your concerns?
> > If there is anything else, any comments would be greatly appreciated.

---

> > > ### Comment · Reviewer_qfrt · 2024-08-13
> > > **Thanks for the response**
> > >
> > > I thank the authors for providing extra experiments. I have updated my score accordingly.

---

> > > > ### Author Response · Authors · 2024-08-13
> > > >
> > > > Thank you for your positive feedback! We appreciate your review and support.

---

### Official Review · Reviewer_PtuD · 2024-07-13

**Soundness:** 4
**Presentation:** 3
**Contribution:** 4
**Rating:** 8
**Confidence:** 2

**Summary:**

The authors present a theoretical analysis of the linear probing and fine-tuning framework based on neural tangent theory, supported by experiments with transformer-based models on natural language processing benchmarks. Their analysis decomposes the NTK matrix into the FT-effective component and the pre-train-effective component, demonstrating an approximation between the fine-tuning matrix and the LoRA matrix.

**Strengths:**

1. The paper is very well written, with detailed technical content. The authors have provided enough mathematical details to support their theoretical analysis. The proofs for theorems and corollaries are detailed delineated in the appendix.

2. A timely topic that bridges established theory and recent advancements in natural language processing.

3. The authors have provided comprehensive benchmarks to support the claims of the paper.

**Weaknesses:**

Although the experiments included an exhaustive list of benchmarks, I noticed that you only implemented one transformer model in your setup. I believe it would be more effective to include multiple types of transformer-based models to better illustrate the practical advantages of your approach.

**Questions:**

Minors: In Figures 3 and 11, it would be helpful to explicitly indicate the purpose of the shaded area in the caption to enhance readability.

**Limitations:**

The authors have made a clear statement about the limitations of the proposed analysis.

---

> ### Author Rebuttal · Authors · 2024-08-04
>
> Thank you very much for taking the time and effort to review our paper. We appreciate your valuable suggestions for improvements.
> > W1: Although the experiments included an exhaustive list of benchmarks, I noticed that you only implemented one transformer model in your setup. I believe it would be more effective to include multiple types of transformer-based models to better illustrate the practical advantages of your approach.
>
> A: We show the results of the T5 model [1] below.
>
> In the main text, we conducted experiments using the RoBERTa model, which is an encoder model. To provide a broader perspective, we included experiments with the T5 model, an encoder-decoder model.
>
> The results from the T5 experiments demonstrate the following:
> 1. Smaller feature changes in LP-FT.
> 1. A significant increase in classifier weight norms during LP.
> 1. Effectiveness of LP-FT.
>
> These findings further support the validity of our analysis presented in the main text.
>
> ### Norm of the feature difference from the pre-trained T5 model
> | Dataset | FT           | LoRA         | LP-FT        | LP-LoRA      |
> |---------|--------------|--------------|--------------|--------------|
> | CB      | 2.77×10¹     | 1.72×10¹     | 9.51×10⁰     | 6.36×10⁰     |
> | RTE     | 1.46×10¹     | 1.43×10¹     | 1.18×10¹     | 7.46×10⁰     |
>
> ### Classifier weight norms of T5 model
> | Dataset | Pretrain       | FT             | LoRA           | LP             | LP-FT          | LP-LoRA        |
> |---------|----------------|----------------|----------------|----------------|----------------|----------------|
> | CB      | 9.97×10⁻¹      | 1.36×10⁰       | 1.51×10⁰       | 1.21×10¹       | 1.21×10¹       | 1.23×10¹       |
> | RTE     | 8.24×10⁻¹      | 1.01×10⁰       | 1.83×10⁰       | 3.69×10⁰       | 1.08×10¹       | 1.14×10¹       |
>
>
> ### Test accuracy of T5 model
> | Dataset | LP             | FT             | LoRA           | LP-FT          | LP-LoRA        |
> |---------|----------------|----------------|----------------|----------------|----------------|
> | CB      | 74.40 ± 2.73   | 82.14 ± 3.09   | **84.52 ± 7.22**   | **84.52 ± 2.06**   | 81.55 ± 1.03   |
> | RTE     | 58.00 ± 1.46   | 73.89 ± 3.28   | **76.17 ± 0.96**   | 75.09 ± 1.30   | 74.97 ± 1.85   |
> > Q1: Minors: In Figures 3 and 11, it would be helpful to explicitly indicate the purpose of the shaded area in the caption to enhance readability.
>
> A: Thank you for pointing this out.
> In Fig.3 and 11, we added the following sentence in the current version:
>
> **"Shaded areas represent standard errors."**
>
> [1] Cong Fang, Hangfeng He, Qi Long, and Weijie J. Su. Exploring deep neural networks via
> layer-peeled model: Minority collapse in imbalanced training. Proceedings of the National
> Academy of Sciences, 118(43), October 2021

---

> > ### Comment · Reviewer_PtuD · 2024-08-12
> >
> > Thank the authors' response and the addition of another set of requirements. I am impressed that the paper's findings hold across various transformer architectures. I adjusted my rating and recommended a strong accept for the paper.

---

> > > ### Author Response · Authors · 2024-08-12
> > >
> > > Thank you for your positive feedback and reevaluation. We appreciate you taking the time to review our paper!

---

### Official Review · Reviewer_R1ei · 2024-07-13

**Soundness:** 3
**Presentation:** 2
**Contribution:** 4
**Rating:** 7
**Confidence:** 4

**Summary:**

This paper studies the dynamics of Linear probing and fine tuning by means of NTK theory. The authors provide a connection between the  Frobenius norm of  linear probing weights and FT-effective component of the NTK matrix.

**Strengths:**

The NTK  sees the model as an  Gaussian process, making it a  powerful tool for analysis of neural networks’ convergence and generalization. This paper  derives an interesting connection between NTK  theory and  fine tuning language models. The NTK is decomposed into  pre-trained and fine tuned terms.  Feature distortion theory is employed to interpret performance of FT.

**Weaknesses:**

-The organization of  material   can be improved. for instance, you  used the linear model $f(x)$ in Proposition 1, however it is introduced later in Definition 1.

-Table 2 is very uninformative and the numbers are not in line, making it hard to read. please provide  precise description of each table and  figure in the text.

-Same for Fig.3, some numbers are floating on it and the message of the figure is not clear.

please  further enhance the presentation of  results,  I found it hard to  infer  your  achievements from the numerical results section.

**Questions:**

In Fig.3, What does “number of feature difference”? please provide the definition.

-How did you calculate the NTK matrix for numerical simulations?  There is a body of literature for only  approximation of NTK for transformers.

-Did you use empirical NTK for simulations? if yes, then can you say if your results hold in the finite width regime too?

**Limitations:**

NTK theory in general holds in the infinite width limit. Further discussions are needed to adapt the methods in this paper to finite regime.

---

> ### Author Rebuttal · Authors · 2024-08-04
>
> Thank you very much for taking the time and effort to review our paper. We appreciate your valuable advice and suggestions for improvements.
>
> > W1: The organization of material can be improved. for instance, you used the linear model  in Proposition 1, however it is introduced later in Definition 1.
>
> A: The model in Proposition 1 is defined on line 136 in Section 4.1. That is, the linear model in Defition 1 is different from one in Proposition 1. However, since this is confusing, we changed the definition of the linear model in Definition 1. We now define the linear model as $f_{\text{linear}}(x) = VBx + b$, i.e., $\phi(x)=Bx$, which is a special case of the general model form $f(x) = V\phi(x) + b$ defined on line 136 and used in Proposition 1.
> > W2: Table 2 is very uninformative and the numbers are not in line, making it hard to read. please provide precise description of each table and figure in the text.
>
> > W3: Same for Fig.3, some numbers are floating on it and the message of the figure is not clear.
>
> A: Thank you for pointing out the formatting issues and the need for clearer descriptions. We recognize the importance of clear and precise presentation and will make appropriate revisions.
>
> ### Formatting Issues in Table 2 and Figure 3
>
> I understand that "293" in Table 2 and "328" in Figure 3, which reference line numbers in the preprint version, have caused confusion. We will ensure these numbers are removed in the camera-ready version of the manuscript to prevent any misinterpretation.
>
> ### Interpretation of Table 2
>
> - The table shows that the FT-effective component outperforms the Pre-train effective component in terms of rank and kernel regression accuracy. This suggests that the FT-effective component has greater expressiveness.
> - The superior performance of LP-FT over the standard method is indicated by higher kernel regression accuracy. The Frobenius norm and the FT ratio suggest that the significant contribution from the FT-component is the reason for this.
>
> We have identified a typo in the Frobenius norm of the LP-LoRA, which should be 15.1, not 1.51. We will correct this in the camera-ready version.
>
> ### Interpretation of Figure 3
> - Figure 3 shows the inverse relationship between the norm of the classifier weight and the norm of the feature difference. This suggests that the large classifier weight norm in LP-FT reduces the feature changes.
> - The lines in the figure represent the mean values, while the shaded areas indicate the standard errors. We will clarify this point in the caption in the camera-ready version of the paper.
>
> ### Conclusion from the numerical results section
> Overall, we validate the following three points in the numerical results section:
> 1. The changes in features during training are smaller in LP-FT than in FT, and the norms of the classifier significantly increase during LP.
> 1. The FT-effective component of the NTK matrix more effectively captures the input data than the pre-train-effective component and is more pronounced in LP-FT than FT.
> 1. A large classifier weight norm reduces the feature change during training, and its negative effects on calibration can be improved by temperature scaling.
>
> > Q1: In Fig.3, What does “number of feature difference”? please provide the definition.
>
> A: That is not **"number"**, is **"norm"**. We understand this might be confusing, and we will change the font size of the figure caption to make it a little more understandable.
>
> In Figure 3, we measure the L2 norm of the difference between features extracted from the trained model and those extracted from the pretrained model. Specifically, for a pre-trained feature extractor $\phi_0$, a trained feature extractor $\phi$, and training examples $x_i (i=1,\cdots N)$, we measure $\frac{1}{N}\sum_{i=1}^N \| \phi(x_i) - \phi_0(x_i)\|_2$.
>
> > Q2: How did you calculate the NTK matrix for numerical simulations? There is a body of literature for only approximation of NTK for transformers.
>
> A: We describe these points in section 7.3.3 in the Appendix.
>
> We separately calculated the pre-train-effective and FT-effective components of the NTK matrix. Following the methodology by Malladi et al. [1], we used functorch and forward-mode auto-differentiation for these calculations. To reduce computational costs, we randomly selected 10% of the parameters from the word embedding matrix for derivative calculations. For datasets with more than 250 samples, we used a subset of 250 randomly selected samples to compute the NTK matrix.
>
> > Q3: Did you use empirical NTK for simulations? if yes, then can you say if your results hold in the finite width regime too?
>
> > L1: NTK theory in general holds in the infinite width limit. Further discussions are needed to adapt the methods in this paper to finite regime.
>
> A: Yes, we used empirical NTK in our experiments.
>
> We assume that the fine-tuning dynamics of large Transformer models is explained with NTK. Although Transformer models have a finite number of parameters, the number is extremely large, and changes in parameters during fine-tuning are smaller compared to standard training.
>
> The empirical NTK has been successfully used to analyze fine-tuning in the finite width regime in previous studies [1, 2]. Therefore, we applied the NTK to analyze the LP-FT method with Transformer models.
>
> [1] Sadhika Malladi, Alexander Wettig, Dingli Yu, Danqi Chen, and Sanjeev Arora. A kernel-based view of language model fine-tuning. In International Conference on Machine Learning, pages 23610–23641. PMLR, 2023.
>
> [2] Alexander Wei, Wei Hu, and Jacob Steinhardt. More than a toy: Random matrix models predict how real-world neural representations generalize. In Proceedings of the 39th International Conference on Machine Learning, 2022.

---

> > ### Comment · Reviewer_R1ei · 2024-08-12
> >
> > Thank you  for responding to my concerns, I increased my rating to 7.

---

> > > ### Author Response · Authors · 2024-08-12
> > >
> > > Thank you for reconsidering our work. We appreciate your feedback and support!

---

### Official Review · Reviewer_zDRZ · 2024-07-15

**Soundness:** 3
**Presentation:** 3
**Contribution:** 4
**Rating:** 7
**Confidence:** 5

**Summary:**

The paper presents a novel application of neural tangent kernel (NTK) theory to analyze the training dynamics of the linear probing then fine-tuning (LP-FT) method for large language models, demonstrating its effectiveness and extending the analysis to include the low-rank adaptation (LoRA) method.

**Strengths:**

The paper stands out for its originality by applying NTK theory to the LP-FT method and extending it to include an analysis of LoRA. It's strong in terms of quality because it combines solid theoretical work with thorough experiments to back up the claims. The writing is clear and well-organized, making it easy to follow the arguments and findings. The significance of the research is evident in its potential impact on fine-tuning practices, not just in NLP but across various fields that use large pre-trained models. Overall, this paper makes a valuable contribution to advancing our understanding and methods in transfer learning.

**Weaknesses:**

One weakness of the paper is that while it presents a solid theoretical foundation and empirical evidence, it could benefit from more detailed explanations in certain sections, particularly the experimental setup and hyperparameter tuning. This would enhance transparency and reproducibility. Additionally, the paper could further strengthen its claims by including a broader range of experiments across different domains beyond NLP. Another point for improvement is the discussion of the impact and practical applications of the findings, which could be expanded to provide more actionable insights for practitioners. Lastly, while the paper briefly mentions temperature scaling to address calibration issues, a deeper exploration of this aspect could provide more comprehensive recommendations for improving model performance.

**Questions:**

Can you expand on the practical applications of your findings? More specific examples of how practitioners can apply your recommendations in real-world scenarios would enhance the practical value of your work.

**Limitations:**

he paper does not discuss the computational resources required for implementing LP-FT and NTK analysis. Providing information on the computational cost and efficiency of the proposed methods would help readers understand the practical feasibility of adopting these techniques.

---

> ### Author Rebuttal · Authors · 2024-08-04
>
> Thank you very much for taking the time and effort to review our paper. We appreciate your valuable advice and suggestions for improvements.
> > W1: One weakness of the paper is that while it presents a solid theoretical foundation and empirical evidence, it could benefit from more detailed explanations in certain sections, particularly the experimental setup and hyperparameter tuning. This would enhance transparency and reproducibility.
>
> A: We have included detailed descriptions of the experimental setup and hyperparameter tuning in Section 7.3 of the Appendix. Additionally, we have made our implementation available on GitHub, though we cannot provide the specific URL at this time due to the anonymity of the review process. Our implementation uses the HuggingFace Transformers library and AdapterHub. For hyperparameter tuning, we conducted a grid search on the validation set. These details are intended to enhance transparency and reproducibility. We agree that transparency and reproducibility are crucial for academic papers.
> > W2: Additionally, the paper could further strengthen its claims by including a broader range of experiments across different domains beyond NLP.
>
> A: The effectiveness of LP-FT in the vision domain has already been validated by Kumar et al. [1], the original authors of LP-FT. Our focus is on applying LP-FT to fine-tuning language models, addressing the current demand in the NLP domain. This is why our experiments are limited to the NLP domain.
> > W4: Lastly, while the paper briefly mentions temperature scaling to address calibration issues, a deeper exploration of this aspect could provide more comprehensive recommendations for improving model performance.
>
> A: We conducted additional experiments of temperature scaling using 8 datasets from the SuperGLUE and GLUE benchmarks, as shown in Tables 14 and 15 in the Appendix. These experiments demonstrate that temperature scaling significantly improves ECE and MCE, particularly with LP-FT and LP-LoRA. We agree that a deeper exploration of temperature scaling to enhance calibration is crucial for practical applications.
> > W3: Another point for improvement is the discussion of the impact and practical applications of the findings, which could be expanded to provide more actionable insights for practitioners.
>
> > Q1: Can you expand on the practical applications of your findings? More specific examples of how practitioners can apply your recommendations in real-world scenarios would enhance the practical value of your work.
>
> A: Our findings have two key practical applications:
> 1. LP-FT for language models: LP-FT is an effective method for fine-tuning language models, as it minimizes changes to valuable pre-trained features.
> 1. Enhancing calibration: We recommend combining LP-FT with temperature scaling to improve calibration.
>
> To demonstrate practical effectiveness, we conducted experiments using the PubMed 20k medical dataset [2], as detailed in Section 7.4.7 and Table 16 of the Appendix. This example provides concrete scenarios for practitioners to apply LP-FT in real-world settings.
>
> [1] Ananya Kumar, Aditi Raghunathan, Robbie Matthew Jones, Tengyu Ma, and Percy Liang. Fine-tuning can distort pretrained features and underperform out-of-distribution. In International Conference on Learning Representations, 2022.
>
> [2] Franck Dernoncourt and Ji Young Lee. Pubmed 200k rct: a dataset for sequential sentence classification in medical abstracts, 2017.

---

> ### Comment · Reviewer_zDRZ · 2024-08-14
> **LGTM**
>
> LGTM, I will keep 7 for now.

---

> > ### Author Response · Authors · 2024-08-14
> >
> > Thank you for reviewing our rebuttal. We are glad that we were able to address your concerns. We appreciate your support.

---

### Author Response · Authors · 2024-08-10
**Confirmation on Addressed Concerns**

Dear AC and reviewers;

We  are concerned because we have not yet received a response from the reviewers and have heard that discussions between reviewers and authors have already started on other papers. We have responded to all of the reviewers' perfectly and additional experiments have supported our theory. Therefore, we believe that all of the reviewer's concerns have been addressed.

---

### Decision · Program_Chairs · 2024-09-25

**Decision:**

Accept (poster)

**Comment:**

This paper uses NTK theory to analyze the training dynamics of the two-stage “linear probing then fine-tuning” (LP-FT) method, and tries to explain why LP-FT consistently outperforms LP or FT alone for both ID and OOD data. This analysis decomposes the NTK matrix into two components, highlighting the importance of the linear head norm alongside the prediction accuracy at the start of the FT stage.

The reviewers agree that this paper presents a solid theoretical contribution. The empirical evidence adds to the contribution of this paper.

The concerns mentioned by the reviewers are mostly about the presentations. A reviewer asked for experiments on additional models, and the results on T5 model were added during the discussion period. The same applies to SQuAD, which also strengthened the generalizability of the finding.